# Small-molecule-induced ERBB4 activation to treat heart failure

Julie M. T. Cools [1,12], Bo K. Goovaerts[1,12], Eline Feyen [1,12], Siel Van den Bogaert[1], Yile Fu[2], Céline Civati[1], Jens Van fraeyenhove[1], Michiel R. L. Tubeeckx[1], Jasper Ott[3], Long Nguyen[4,5], Eike M. Wülfers [6,7], Benji Van Berlo[1], Antoine A. F. De Vries [8], Nele Vandersickel [6], Daniël A. Pijnappels [8], Dominique Audenaert[4,5], H. Llewelyn Roderick [2], Hans De Winter [9], Gilles W. De Keulenaer[1,10,13] & Vincent F. M. Segers [1,11,13] ✉

Heart failure is a common and deadly disease requiring new treatments. The neuregulin-1/ERBB4 pathway offers cardioprotective benefits, but using recombinant neuregulin-1 as therapy has limitations due to the need for intravenous delivery and lack of receptor specificity. We hypothesize that small-molecule activation of ERBB4 could protect against heart damage and fibrosis. To test this, we conduct a screening of 10,240 compounds and identify eight structurally similar ones (EF-1 to EF-8) that induce ERBB4 dimerization, with EF-1 being the most effective. EF-1 reduces cell death and hypertrophy in cardiomyocytes and decreases collagen production in cardiac fibroblasts in an ERBB4-dependent manner. In wild-type mice, EF-1 inhibits angiotensin-II-induced fibrosis in males and females and reduces heart damage caused by doxorubicin and myocardial infarction in females, but not in Erbb4-null mice. This study shows that small-molecule ERBB4 activation is feasible and may lead to a novel class of drugs for treating heart failure.

Over 20 years ago, it was reported that humans treated with anti-ERBB2 antibodies for ERBB2-positive breast cancer, and mice with conditional post-natal deletion of *Nrg1*, *Erbb2*, or *Erbb4* tyrosine kinase receptors, develop cardiomyopathy[1–6]. Since then, the ERBB system has been recognized as a cardioprotective system. The ERBB system regulates cardiac embryonic development, preserves normal cardiac function, and is activated during cardiac overload or injury as part of a compensatory mechanism[7–9]. The protective effects of the ERBB system are attributed to various mechanisms, including regenerative, pro-survival, anti-fibrotic, and anti-inflammatory mechanisms. These

mechanisms act on various cardiac cell types, such as cardiomyocytes, endothelial cells, fibroblasts, and inflammatory cells, all of which express one or more ERBB subtypes[10–14].

The ERBB receptor system, with its attractive biological profile, is a promising albeit complex therapeutic target[15]. Multiple endogenous peptide growth factors have been identified as ligands. Each of these ligands binds to one or more ERBB receptors, except for ERBB2 which only functions as a dimerization partner, and operates as either a full or partial agonist[16]. Upon activation, ERBB4 receptors can form homo-dimers or heterodimers with ERBB2 or ERBB3, which leads to different

[1]Laboratory of PhysioPharmacology, University of Antwerp, Antwerp, Belgium. [2]Laboratory of Experimental Cardiology, Department of Cardiovascular Sciences, KU Leuven, Leuven, Belgium. [3]Laboratory of Cell Biology and Histology, University of Antwerp, Antwerp, Belgium. [4]Screening Core, VIB, Ghent, Belgium. [5]Centre for Bioassay Development and Screening (C-BIOS), Ghent University, Ghent, Belgium. [6]Department of Physics and Astronomy, Ghent University, Ghent, Belgium. [7]Institute for Experimental Cardiovascular Medicine, University Heart Center Freiburg - Bad Krozingen, Freiburg im Breisbau, Germany and Faculty of Medicine, University of Freiburg, Freiburg im Breisgau, Germany. [8]Laboratory of Experimental Cardiology, Leiden University Medical Center, Leiden, the Netherlands. [9]Laboratory of Medicinal Chemistry, University of Antwerp, Antwerp, Belgium. [10]Department of Cardiology, ZNA Hospital, Antwerp, Belgium. [11]Department of Cardiology, University Hospital Antwerp, Antwerp, Belgium. [12]These authors contributed equally: Julie M. T. Cools, Bo K. Goovaerts, Eline Feyen. [13]These authors jointly supervised this work: Gilles W. De Keulenaer, Vincent F. M. Segers. ✉e-mail: vincent.segers@uantwerpen.be

post-receptor signaling events and biological effects. Many researchers studying the activation of this complex system have used variants of the NRG1 protein, one of the system's most active ligands[17]. In animal experiments, NRG1 application improves various aspects of cardiac disease, including heart failure mortality, systolic and diastolic dysfunction, myocardial infarct size, and myocardial fibrosis[9,18–21]. Although progress has been made, these effects of NRG1 remain to be sufficiently reproduced in clinical trials[22–24].

Use of recombinant NRG1 (rNRG1) as a therapeutic has three major disadvantages. First, it has a short half-life of 15 min. Second, it requires intravenous administration, which limits its applicability in chronic diseases such as heart failure. Third, NRG1 is a non-selective agonist of the ERBB system, binding to both ERBB3 and ERBB4 receptors, and thus it may activate ERBB3 receptors in ERBB2-overexpressing tumor cells, thereby enhancing the formation of ERBB3/ERBB2 complexes, which could induce or accelerate cancer growth[25,26].

Although the identification of small molecules that induce cell surface receptor dimerization has been recognized to be challenging[8], the hypothesis of this study is that small molecule-induced activation of the ERBB system is feasible. Here, we screened for compounds that induce ERBB4 dimerization, for several reasons. First, most cardioprotective effects of the ERBB system are dependent on the activation of ERBB4[9,15,27]. Second, selective activation of ERBB4 preferentially over ERBB3 has a much safer oncological profile[28]. Specifically, previous studies with engineered bivalent NRG1 that activates the ERBB system by forced ERBB4 homodimerization, reproduced the anti-apoptotic effects of NRG1 in cardiomyocytes, without inducing growth of cancer cells[29]. For this reason, we performed a high-throughput screen (HTS) for ERBB4 ligands using an assay that detects ERBB4 homodimerization upon ligand binding. Here, we report the identification of small molecules that induce ERBB4 homodimerization, with compound EF-1 being the most potent and promising compound. EF-1 activates similar pathways as NRG1, is cardioprotective in vitro and in vivo via the ErbB4 receptor.

## Results

### Identification of small molecule ERBB4 agonists

To identify small-molecule agonists of the ERBB4 receptor, an HTS was performed with a Pharmacological Diversity Set of 10,240 synthetic molecules using an ERBB4/ERBB4 dimerization assay (Supplementary Table 1). A primary screen followed by a confirmation screen resulted in 62 hit compounds (Fig. 1a). To reduce, confirm and cluster the compounds, maximum common substructure (MCS) analysis was performed resulting in the identification of 368 unique MCSs within the 62 confirmed hits. The outcome of a k-means clustering on these 368 MCSs resulted in 10 unique cluster centers, of which cluster center 4 showed a 458-fold enrichment compared to the Enamine reference library (Supplementary Fig. 1a). Cluster center 4 consisted of 61 individual cluster members, 3 of which provided a significant enrichment compared to the reference library.

The common MCS derived from these 3 MCSs resulted in a pattern that gives 109-fold enrichment (Supplementary Fig. 1b). Four of the compounds of the confirmed hit list contained this pharmacophore, and 16 compounds contained analogous MCS patterns. We generated a dose-response curve (DRC) of these 20 compounds. Only the 4 hit compounds containing the common pharmacophore (named EF-1, EF-2, EF-3, and EF-4) showed a reliable and reproducible DRC with half maximal effective concentration ($EC_{50}$) values in the micromolar range and a maximum response ($E_{max}$) that varied from $10.9 \pm 2.2\%$ to $27.9 \pm 4.8\%$, relative to NRG1 (Fig. 1b, f–h). We screened an additional 111 compounds containing the pharmacophore (selected from the Enamine library), which resulted in 2 additional hits (EF-5 and EF-6) that induced ERBB4 dimerization in a concentration-dependent manner (Fig. 1i, j).

Next, a random-forest machine-learning model was developed using Morgan fingerprints of active and inactive molecules based on the results of the 20 selected hit compounds and the results of the additional screening of 111 analogues. Applying this machine-learning model on the Enamine library led to 34 additional compounds that were predicted to be active. Experimental validation of these 34 predictions resulted in 2 additional compounds (EF-7 and EF-8) that induced ERBB4 dimerization in a concentration-dependent manner (Fig. 1k, l). Apart from the 8 active pharmacophore-containing ERBB4 agonists, we selected 1 of the compounds that did not activate ERBB4 but contained the same pharmacophore (hereafter named NA-1, Fig. 1m) to be included in several experiments as a control. The chemical structures and molecular weights (MW) of the 8 selected hit compounds and NA-1 are shown in Supplementary Fig. 2.

We selected EF-1 for further studies because it showed the highest potency ($EC_{50} = 10.5 \pm 4.5 \times 10^{-6}$ M) and efficacy ($E_{max} = 27.9 \pm 4.8\%$ of the effect of NRG1; Fig. 1b) in the ERBB4/ERBB4 dimerization assay. Since ERBB2 is a preferred heterodimerization partner of ERBB4, and because NRG1 can also bind to ERBB3 to induce ERBB2/ERBB3 dimerization, we evaluated the effects of EF-1 on ERBB2/ERBB4 and ERBB2/ERBB3 dimerization. Although the potency of EF-1 in the ERBB2/ERBB4 dimerization assay was low ($EC_{50} > 32 \mu M$), its efficacy at the highest concentration was like the natural ligand NRG1 (Fig. 1c). The stimulatory effect of EF-1 on ERBB2/ERBB3 heterodimerization was lower compared to ERBB4 homodimerization (Fig. 1d), indicating a preferential but not exclusive activation of ERBB4. We also noted that EF-1 activated ERBB1/ERBB1 dimers, though this effect required again a higher concentration ($EC50 > 32 \mu M$, Fig. 1e).

To evaluate whether EF-1 binds to a similar binding pocket than NRG1, we performed a cell-based competition assay. A flow cytometry–based fluorescence competition assay showed that NRG1 as a positive control (ctrl) reduced binding of fluorescent NRG1 (F-NRG1) to the ERBB4 receptor with a half maximal inhibitory concentration ($IC_{50}$) of 32 nM (Supplementary Fig. 3a). EF-1, however, did not reduce binding of F-NRG1 (Supplementary Fig. 3a), indicating that EF-1 and NRG1 bind to different sites. To evaluate whether EF-1 could influence the effect of NRG1 on ERBB4 dimerization, the ERBB4/ERBB4 dimerization cell line was used to generate DRCs of NRG1 in the presence of 0 to $32 \mu M$ of EF-1. EF-1 dose-dependently and significantly potentiated NRG1-induced ERBB4 homodimerization, by 299.5% at a concentration of $32 \mu M$ (Supplementary Fig. 3b).

### EF-1 and NRG1 activate similar downstream signaling pathways

Canonical pathways activated by NRG1 in cardiomyocytes are the phosphatidylinositol 3-kinase (PI3K)/protein kinase B (Akt) and mitogen-activated protein kinase (MAPK)/extracellular signal-regulated kinase (ERK) pathways[30–32]. Phosphorylation of Akt and ERK1/2 was assessed by western blot analysis in conditionally immortalized rat atrial myocytes (iAMs)[33]. Compared to cardiomyocytes derived from induced pluripotent stem cells, iAMs are easier to culture and show a more complete and reliable cardiomyogenic differentiation[33]. As shown in Fig. 2a, NRG1 showed a significant increase in phosphorylated Akt levels after already 15 minutes. This phosphorylation decreased over time. Additionally, NRG1 showed a time-dependent phosphorylation of ERK1/2 (Fig. 2a). This experiment was repeated with EF-1 on both pathways. Compound EF-1 induced a time-dependent phosphorylation of Akt (Fig. 2b) in cardiomyogenically differentiated iAMs. EF-1 also induced a transient phosphorylation of ERK1/2 (Fig. 2b).

To gain insight into downstream signaling pathways and cell responses activated by small-molecule ERBB4 agonists, we performed bulk RNA sequencing on cardiomyogenically differentiated iAMs and human cardiac fibroblasts (HCF), stimulated with either EF-1, NRG1, or vehicle (Veh). In iAMs, EF-1 significantly upregulated 354 genes and significantly downregulated 1120 genes compared

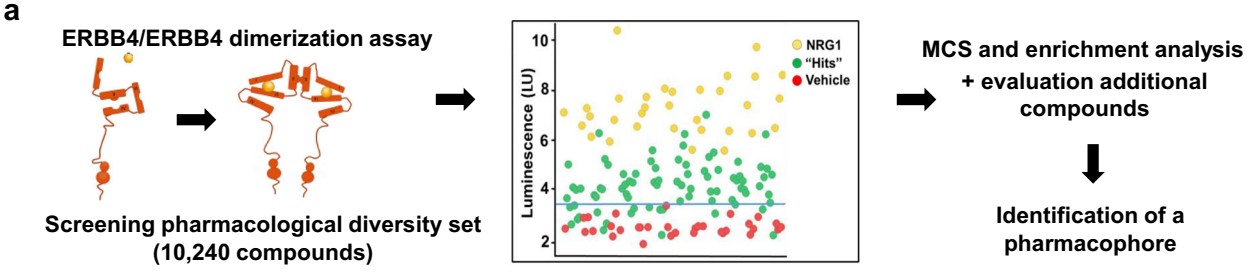

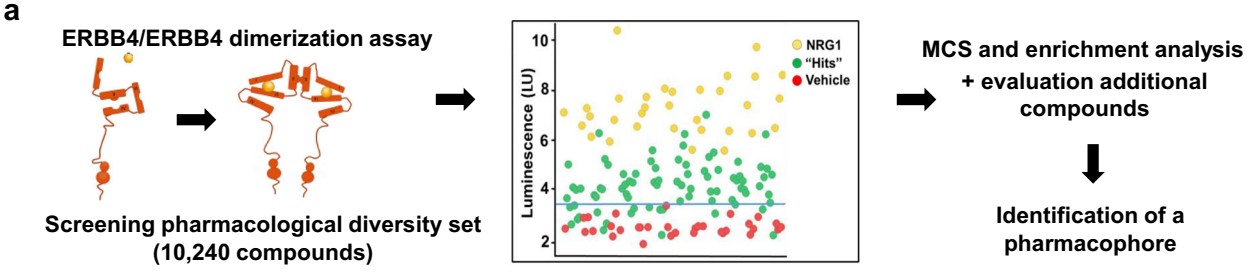

**Fig. 1 | High-throughput screening to identify small-molecule ERBB4 agonists.**
**a** An ERBB4/ERBB4 dimerization assay was used to screen 10,240 pharmacologically diverse compounds for their ability to induce ERBB4 homodimerization. The confirmation screen resulted in 62 hits (green dots) with a luminescence signal above threshold (light blue line, $3.5 \times 10^6$ LU). MCS analysis and enrichment analysis of the confirmed hits resulted in the identification of a pharmacophore. After the screening and evaluation of additional compounds, 8 candidate ERBB4 agonists were selected. Yellow dots, positive control (NRG1); red dots, negative control (Veh); green dots, hit compounds. DRC of the most potent compound EF-1 in **b** the ERBB4/ERBB4 dimerization assay ($n = 4$ biological replicates for every concentration), **c** the ERBB2/ERBB4 dimerization assay ($n = 8$ biological replicates for every concentration), **d** the ERBB2/ERBB3 dimerization assay ($n = 6$ biological replicates for every concentration) and **e** the ERBB1/ERBB1 dimerization assay ($n = 6$ biological replicates for every concentration). **f**–**l** DRCs of the 7 other selected hit compounds ($n = 3$ for (**g**), $n = 4$ for (**f**, **i**–**l**) and $n = 5$ for (**h**) biological replicates for every concentration) and (**m**) a non-active compound NA-1 on the ERBB4/ERBB4 dimerization assay ($n = 2$ biological replicates for every concentration). The signal in the dimerization assays is plotted relative to the signal obtained with $0.1\,\mu$M NRG1 (**b**–**d**, **f**–**m**) or $0.1\,$g/mL EGF (**e**). All data are represented as mean ± SD. Source data are provided as a Source Data file. DRC dose response curve, EGF epidermal growth factor, LU luminescence, MCS maximum common substructure, NA non-active pharmacophore-containing compound, NRG1 neuregulin-1, Veh vehicle.

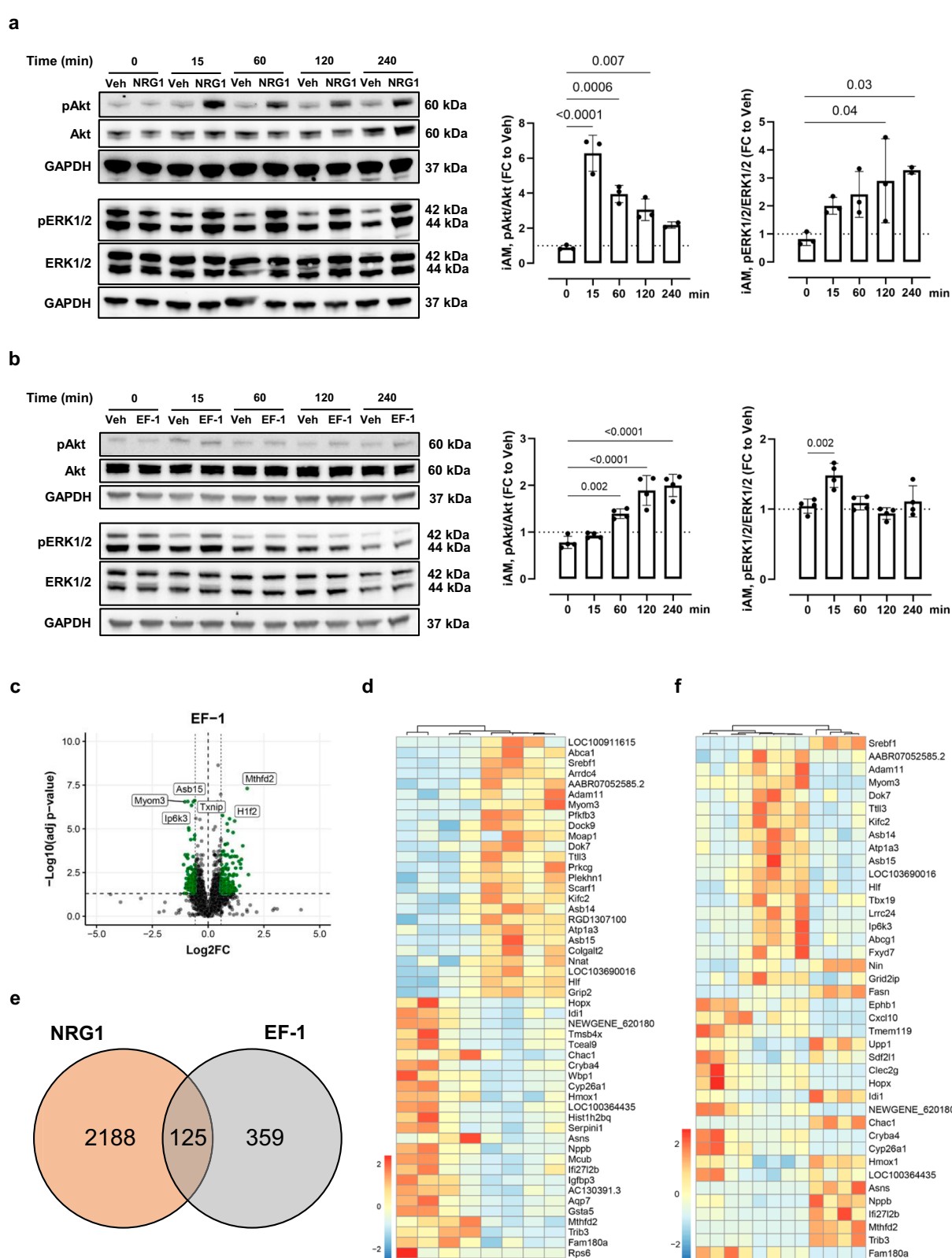

to Veh (log$_2$-fold change ≥0.58, $P$ adj <0.05) (Fig. 2c). The top 50 significant differentially expressed genes (DEGs) between EF-1 and the Veh group are shown in Fig. 2d. Although there is a substantial overlap in DEGs between NRG1 and EF-1 (Fig. 2e), many DEGs induced by NRG1 and EF-1 differ, which is also reflected in a cluster analysis (Fig. 2f).

In HCFs, EF-1 exhibited a significant upregulation of 336 genes and a significant downregulation of 507 genes compared to Veh (log2-fold change >0, $P$ adj <0.05) (Fig. 3a). Figure 3b displays the top 50 significantly DEGs between EF-1 and the control group. Additionally, nearly two-thirds of the DEGs influenced by NRG1 are shared with DEGs influenced by EF-1 (Fig. 3c, d). In HCFs, the transforming growth factor-

**Fig. 2 | EF-1 and NRG1 activate similar downstream signaling pathways in cultured cardiomyocytes.** Representative western blot images showing the effect of (**a**) NRG1 (10 nM) and (**b**) EF-1 (32 μM) on pAkt/Akt and pERK1/2/ERK1/2 pathways in iAMs stimulated for different times. Bar graphs represent the average effects of phosphor-markers normalized to total levels of each protein, relative to vehicle of the same time point; $n = 3$ (**a**) or $n = 4$ (**b**) biological replicates ± SD; one-way ANOVA with Dunnett's multiple comparisons test. The samples derive from the same experiment and the blots were processed in parallel. Despite the subtle changes in the representative blots of EF-1, the results were highly reproducible. The uncropped gels are presented in Supplementary Figs. (a) 7 and (b) 8. **c** Volcano plot showing differential gene expression (two-tailed DESeq2's Wald test, $P$ adj <0.05) of iAMs in response to EF-1 ($n = 4$ biological replicates). Vertical dashed lines indicated log2 fold change ≥0.58. The horizontal dashed line indicates -Log10 ($P$ adj <0.05). **d** Heatmap showing log2 fold change estimates for top DEGs in EF-1-treated iAMs relative to the control group. **e** Venn diagram of overlapping DEGs between the NRG1 and EF-1 group. **f** Heatmap showing log2 fold change estimates for top DEGs in EF-1- or NRG1-treated iAMs relative to the control group. Source data are provided as a Source Data file. Akt protein kinase B, ANOVA analysis of variance, DEG differentially expressed gene, ERK extracellular signal-regulated kinase, FC fold change, GAPDH glyceraldehyde-3-phosphate dehydrogenase, iAM conditionally immortalized rat atrial myocyte, NRG1 neuregulin-1, SD standard deviation, Veh vehicle.

---

β (TGF-β) and MAPK/ERK pathways were both downregulated by EF-1 and by NRG1, but only reaching statistical significance with EF-1 (Fig. 3e). The PI3K/Akt pathway was downregulated in HCFs when stimulated with EF-1 or NRG1, but not statistically significant (Fig. 3e). To confirm the RNA sequencing data, we performed qRT-PCR analysis on two genes from each signaling pathway, i.e. those with the largest decrease in expression induced by both EF-1 and NRG1. All six genes, belonging to either the TGF-β (Fig. 3f), MAPK/ERK (Fig. 3g) or PI3k-Akt pathway (Fig. 3h) showed a significant decrease in expression when HCFs were treated with EF-1 or NRG1.

### EF-1 decreases TGF-β1-induced collagen expression through ERBB4 in vitro

Because NRG1 has been shown to decrease collagen mRNA expression induced by TGF-β1 in fibroblasts, we determined the effects of EF-1 on collagen expression induced by TGF-β1 (10 ng/mL) in HCFs and included an *ERBB4* knockdown experiment to test the role of ERBB4[11,34] We first observed that TGF-β1 significantly increased collagen type 3 alpha 1 (*COL3A1*) mRNA levels in HCFs by 36% (Fig. 4a) and that the expression of *ERBB4* in HCFs was downregulated by 60% by *ERBB4* silencing RNAs (siRNAs) (Fig. 4b). Next, EF-1 dose-dependently decreased *COL3A1* mRNA levels, induced by TGF-β1, up to 50%, while knockdown of *ERBB4* expression significantly attenuated this effect (Fig. 4c). In contrast, NA-1, a compound that contains the pharmacophore but does not induce ERBB4 dimerization, did not affect *COL3A1* expression in HCFs induced by TGF-β1 (Fig. 4c).

### EF-1 reduces cardiomyocyte cell death and hypertrophy in vitro

Because NRG1 has been shown to decrease cardiomyocyte cell death and to attenuate cardiomyocyte hypertrophy[10,35,36], we next evaluated the effects of EF-1 on cardiomyocytes and again included an *ERBB4* knockdown experiment. Cardiotoxicity was induced in iAMs with hydrogen peroxide (H₂O₂). We first observed that exposure of iAMs to 100 μM H₂O₂ increased total cell death to 62% (Fig. 4d) and that *Erbb4* siRNAs reduced *ERBB4* mRNA levels in iAMs by more than 50% (Fig. 4e). Next, EF-1 dose-dependently decreased H₂O₂-induced cardiomyocyte cell death with more than 75% at its highest dose, an effect that was diminished by *Erbb4* knockdown (Fig. 4f). In contrast, NA-1 did not have a significant effect on cardiomyocyte survival (Fig. 4f). The protective effects of EF-1 on cardiomyocyte viability were further confirmed by 2-(4-iodophenyl)-3-(4-nitrophenyl)-5-(2,4-disulfophenyl)-2H-tetrazolium (WST-1) and terminal deoxynucleotidyl transferase dUTP nick end labeling (TUNEL) assay. Treatment with 100 μM H₂O₂ reduced iAM cell viability, as indicated by decreased WST-1 absorbance, but pretreatment with EF-1 significantly increased cell viability (Supplementary Fig. 4a). H₂O₂ also significantly increased the number of TUNEL-positive (apoptotic) cells, and EF-1 also significantly reduced this number (Supplementary Fig. 4b). Finally, Ang II induced cardiomyocyte hypertrophy, as indicated by a significant increase in cross sectional area (CSA), and this effect was dose-dependently attenuated by EF-1 (Fig. 4g).

As shown in Fig. 2b, EF-1, as NRG1, signals through the Akt- and ERK1/2-pathway[37]. To determine if the observed cardiomyocyte survival and anti-hypertrophic effects of EF-1 are Akt- and ERK1/2-pathway related, we repeated the experiments with pre-incubation of iAMs with LY294002 (a PI3K/Akt-inhibitor), and PD98059 (a MEK1/2-inhibitor). We observed that pretreatment with LY294002 blocked the pro-survival effect of EF-1 on iAMs treated with H₂O₂ (Fig. 4h), suggesting that that the PI3K/Akt pathway contributes to the cell survival capacities of EF-1. In addition, we observed that the anti-hypertrophic effect of EF-1 was blocked upon inhibition of the ERK1/2 pathway (Fig. 4i).

### EF-1, as NRG1, did not promote iAM proliferation

Because NRG1 promotes cardiomyocyte proliferation[13,14], although debatable[38], we tested if EF-1 could promote iAM proliferation. We performed an 5-ethynyl-2'-deoxyuridine (EdU)-staining on proliferating iAMs, stimulated with either EF-1 or NRG1, and observed a significant decrease in EdU-positive nuclei in both groups compared to the vehicle group (Supplementary Fig. 4c).

### EF-1 prevents myocardial fibrosis

Before switching to in vivo studies, we evaluated the pharmacological stability of EF-1. EF-1 remained stable in plasma of both human and mouse for at least 6 h (Supplementary Fig. 5a). Moreover, the half-life of EF-1 was >6 h when incubated with human liver microsomes and 15 min when incubated with mouse microsomes (Supplementary Fig. 5b).

Next, we evaluated the effects of EF-1 in a mouse model of AngII-induced myocardial fibrosis[11,39]. The experimental design is shown in Fig. 5a. To evaluate changes in mRNA expression of markers of fibrosis, EF-1 was administered simultaneously with AngII for 1 week in male wild type mice. EF-1 not only significantly reduced *Col1a1* and *Col3a1* mRNA expression induced by AngII (Fig. 5b, c), indicating anti-fibrotic and cardioprotective properties, but also significantly decreased atrial natriuretic peptide (*Nppa*) expression (Fig. 5d), a cardiac stretch marker. To study effects on tissue fibrosis, EF-1 was administered simultaneously with AngII for 4 weeks in male and female wild type mice. Masson's trichrome staining of myocardial tissues showed that EF-1 significantly prevented both interstitial and perivascular fibrosis, induced by AngII (Fig. 5e, f). EF-1 did not significantly alter cardiac dimensions on ultrasound in both sexes (Supplementary Table 2a, b).

To evaluate whether the observed anti-fibrotic effects could be mediated by targets unrelated to ERBB4, we performed a similar experiment with NA-1 (containing the common pharmacophore but without induction of ERBB4 dimerization) in male wild type mice. NA-1 did not significantly prevent interstitial or perivascular AngII-induced fibrosis (Supplementary Fig. 6a). Additionally, we evaluated the effects of EF-1 in Ang II-treated male and female transgenic mice with tamoxifen-induced deletion of *Erbb4* (*Erbb4*-null mice). *Erbb4* deletion was confirmed in the heart by western blot analysis (Supplementary Fig. 6b). As expected, *Erbb4* deletion resulted in a cardiomyopathy phenotype, with a significant increase in left ventricular internal diameter in diastole (LVIDd) and systole (LVIDs), and a fall in fractional shortening (FS) (Supplementary Table 2d).

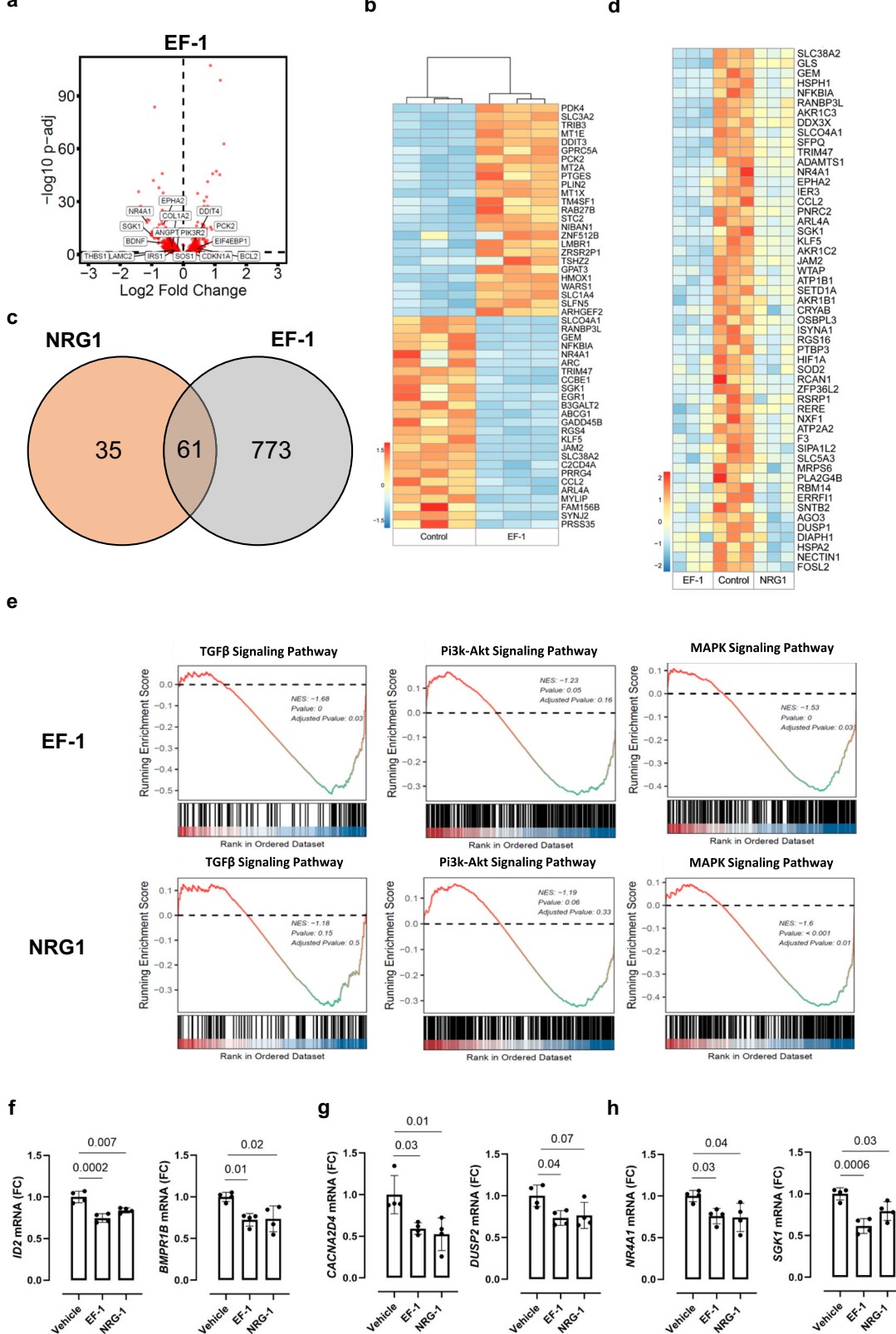

Since the commonly used AngII dose of 1000 ng/kg/day induced 80% mortality in the Erbb4-null mice (Supplementary Fig. 6c), the dose of AngII was lowered to 400 ng/kg/day[40]. EF-1 did not significantly prevent interstitial or perivascular AngII-induced fibrosis in *Erbb4*-null mice indicating that ERBB4 is necessary for the effects of EF-1 on cardiac fibrosis (Fig. 5g). Of note, when *iCAGCre-Erbb4ᶠ/ᶠ* mice were injected with corn oil without tamoxifen, the inhibitory effects of EF-1 on

AngII-induced interstitial and perivascular fibrosis were preserved (Supplementary Fig. 6d).

**EF-1 prevented cardiomyocyte injury in doxorubicin (DOX)-treated mice**

Because ERBB4 activation is known to prevent cell death, we investigated the effects of EF-1 in a mouse model of DOX-induced acute

**Fig. 3 | EF-1 and NRG1 activate similar downstream signaling pathways in human cultured cardiac fibroblasts. a** Volcano plot showing differential gene expression (two-tailed DESeq2's Wald test, *P* adj <0.05) of HCFs in response to EF-1 (*n* = 3 biological replicates). The horizontal dashed line indicates -Log10 (*P* adj <0.05). **b** Heatmap showing log2 fold change estimates for top DEGs in EF-1-treated HCFs relative to the control group. **c** Venn diagram of overlapping DEGs between the NRG1 and EF-1 group. **d** Heatmap showing log2 fold change estimates for top DEGs in EF-1- or NRG1-treated HCFs relative to the control group. **e** GSEA of HCFs with EF-1 or NRG1 for gene signatures of TGF-β, PI3K/Akt and MAPK pathway-regulated genes. *P*-values were adjusted via the Benjamini–Hochberg procedure. **f–h** RT-qPCR on signaling pathway genes in HCFs; normalized FC compared to vehicle (*n* = 4 biological replicates for every value in each group): **f** *ID2* and *BMPR1B*, genes involved in TGFβ signaling, **g** *CACNA2D4* and *DUSP2*, genes involved in

MAPK/ERK signaling and **h** *NR4A1* and *SGK1*, genes involved in PI3k-Akt signaling. All data are represented as mean ± SD, one-way ANOVA with Tukey's multiple comparisons test. Source data are provided as a Source Data file. Akt protein kinase B, ANOVA analysis of variance, BMPR1B bone morphogenetic protein receptor type 1B, CACNA2D4 calcium voltage-gated channel auxiliary subunit alpha-2 delta-4, DEG differentially expressed gene, DUSP2 dual specificity phosphatase 2, ERK extracellular signal-regulated kinase, FC fold change, GSEA gene set enrichment analysis, HCF human cardiac fibroblast, ID2 Inhibitor of DNA Binding 2, MAPK mitogen-activated protein kinase, NES normalized enrichment score, NRG1 neuregulin-1, NR4A1 nuclear receptor subfamily 4 group A member 1, PI3K phosphatidylinositol 3-kinase, RT-qPCR real-time quantitative polymerase chain reaction, SGK1 serum/glucocorticoid-regulated kinase 1, TGF-β transforming growth factor-β.

cardiac toxicity[29,41,42]. The experimental design of this experiment is shown in Fig. 6a. Osmotic minipumps with either EF-1 or Veh were implanted in male and female wild type and female *Erbb4-null* mice. Four days after the start of the EF-1 or Veh treatment, the animals received an intraperitoneal injection of 20 mg/kg DOX followed 3 days later by the measurement of cardiac troponin I (cTnI) plasma levels. EF-1 significantly prevented the DOX-induced increase in circulating cTnI plasma levels in female mice (Fig. 6b), but not in male mice (Fig. 6c). To verify whether the cardioprotective effects of EF-1 in female mice were mediated by ERBB4, we repeated the experiment using female *Erbb4*-null mice and observed that EF-1 did not prevent the increase in cTnI plasma levels in these mice (Fig. 6d).

### EF-1 reduces cardiac remodeling and interstitial fibrosis after MI
In view of the above effects induced by EF-1, and also because NRG1 has been shown to attenuate adverse cardiac remodeling in MI models, we assessed the effects of EF-1 in a murine MI model (Fig. 7a)[27,43,44]. Male and female mice were randomized 1 week after ligating the left anterior descending artery (LAD) to implantation of osmotic minipumps with either EF-1 or Veh. Treatment lasted for 28 days, during which cardiac size and function was evaluated using echocardiography (Supplementary Table 2e, f). Representative echocardiographic images of all groups in female mice are shown in Fig. 7b. In Veh-treated female mice, MI significantly increased left ventricular end-diastolic volume (LVEDV; Fig. 7c) and left ventricular end-systolic volume (LVESV; Fig. 7d) and decreased left ventricular ejection fraction (EF; Fig. 7e). Four-week treatment with EF-1 significantly attenuated the increase in LVEDV and LVESV and resulted in a trend towards an increased EF (*p* = 0.0858). In male mice, EF-1 did not have affect LVEDV, LVESV, or EF (Fig. 7c–e).

Hypertrophy and vascularization in heart tissue were visualized using an isolectin-B4/Wheat Germ Agglutinin (WGA)/4',6-diamidino-2-phenylindole dihydrochloride (DAPI) staining. There was a trend to increase in cardiomyocyte cross-sectional area in the MI/Veh and MI/EF-1 group and a significant increase in cross-sectional area in the MI/EF-1 group, in both male and female mice (Fig. 7f). Additionally, a marked decrease in number of capillaries per area (μm²) of the heart was detected in both MI/Veh and MI/EF-1 groups in female mice, but not in male mice (Fig. 7g). This decrease was not observed when the number of capillaries was normalized to cardiomyocytes, in both sexes (Fig. 7h).

Interstitial fibrosis in the remote myocardium of the infarcted hearts was significantly lower in female mice treated with EF-1, but not in male mice (Fig. 7i, j). EF-1 did not affect scar size in female or male mice (Fig. 7k).

### Discussion
In this study, we aimed to investigate the effects of ERBB4 small-molecule agonists on cardiac function and survival. By combining HTS and chemoinformatics, we identified a series of compounds that consist of a common pharmacophore and that induce dimerization of

the ERBB4 receptor. The most potent compound, EF-1, partially activated downstream signaling pathways that are activated by NRG1, the natural ligand of ERBB4. EF-1 decreased fibroblast collagen production, decreased cardiomyocyte hypertrophy, and improved cardiomyocyte survival in vitro. In vivo, EF-1 mitigated myocardial fibrosis in an AngII model of cardiac fibrosis in both male and female mice, prevented acute cardiomyocyte injury in DOX-treated female mice, and reduced cardiac dilation and cardiac fibrosis in female mice that underwent MI. We showed that the in vitro effects of EF-1 were ERBB4-dependent since they could be abrogated by siRNAs against *ERBB4*. Moreover, the in vivo effects of EF-1 could be abrogated by transgenic deletion of *Erbb4* in mice. Finally, a compound containing the same pharmacophore as EF-1 but without inducing ERBB4 dimerization (NA-1), did not have the same in vitro and in vivo effects as EF-1, further supporting that the effects of EF-1 depend on its ability to induce ERBB4 dimerization.

Previous studies have shown the importance of the NRG1/ERRB4 signaling pathway in cardiac development, physiology, and adaptation during disease[8,45,46]. Based on these studies, rNRG1 has been developed as a potential therapy for chronic heart failure (CHF). Despite encouraging results in phase I and II clinical trials over a decade ago[22,23], no results of phase III trials have been published yet. Due to its short plasma half-life, rNRG1 must be administered by continuous intravenous infusion. Moreover, in the published phase I and II trials[22,23], rNRG1 was administered for 10 consecutive days without follow-up treatment. Although, short-term effects of a brief rNRG1 administration were significant, it seems unlikely that this treatment regimen will provide long-lasting benefits for CHF patients. Therefore, development of small-molecule ERBB4 agonists could result in a more efficacious therapy for CHF patients.

Receptor dimerization is an established mechanism for the initiation of signal transduction, seen in many cell surface receptors[47]. Examples are protein–tyrosine kinase receptors (including ERBB4), the tumor necrosis factor receptor family, protein-serine/threonine kinase receptors, antigen receptors, and members of the cytokine receptor superfamily[47]. Discovery of small molecules with agonist activity on receptor dimerization has been recognized as a challenging endeavor[47]. There are few publications on small molecules capable of disrupting specific protein–protein interactions (antagonists), and the additional requirement for agonists to bind to but also induce dimerization of two receptor molecules makes this aim even more challenging[47]. For instance, considerable effort has been put into the identification of small non-peptide agonists of the erythropoietin receptor, resulting in the identification of weak activators[48]. Screening for compounds activating the related thrombopoietin receptor (TPOR) resulted in identification of a potent activator, called SB394725[49]. Structural studies have shown that SB394725 interacts with the juxtamembrane residues of the transmembrane region of TPOR[50], but whether SB394725 directly induces dimerization of TPOR is unclear.

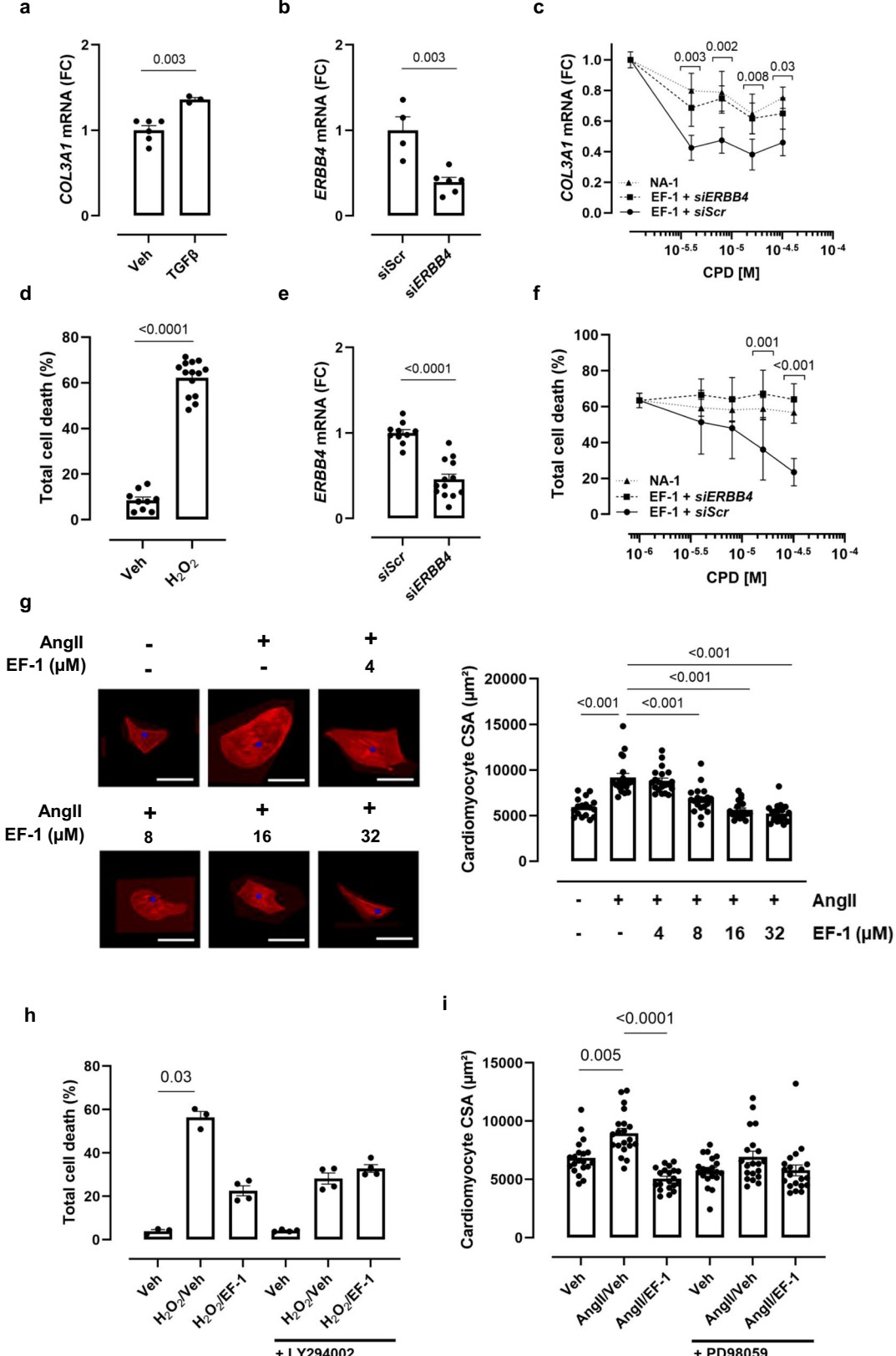

The location of the exact binding site of the hit compounds on ERBB4 is currently unknown, but most likely differs from NRG1, as our data indicate that EF-1 does not compete with NRG1 for binding to U2OS ERBB4/ERBB4 dimerization cells and because EF-1 potentiates the effect of NRG1 on homodimerization of ERBB4. This is consistent with a mechanism of action based on ago-allosteric modulation of ERBB4 receptor dimerization[51]. Allosteric activation of ERBB4 and potentiation of the effects of NRG1 could partially explain the remarkable biological effects of EF-1 in vitro and in vivo, despite its potency being lower than that of NRG1. Nevertheless, identification of the binding site of the small molecules could facilitate computational screening of novel small molecules and in silico optimization. Binding

**Fig. 4 | EF-1 decreases collagen production in fibroblasts, and decreases cardiomyocyte cell death and hypertrophy. a** Effect of TGF-β1 on *COL3A1* mRNA expression in HCFs (*n* = 6 biological replicates for Vehicle group and *n* = 3 biological replicates for TGFβ group). **b** ERBB4 knockdown efficiency of siRNAs against *ERBB4* compared to control (i.e. scrambled) siRNAs in HCFs (*n* = 4 biological replicates for *SiScr* group and *n* = 6 biological replicates for *SiERBB4* group). **c** *COL3A1* mRNA expression after stimulation with EF-1 (4–32 μM) and TGF-β1 in the presence of scrambled (full line with full circle; *n* = 3 biological replicates for every concentration) or *ERBB4*-specific siRNAs (dashed line with square; *n* = 6 biological replicates for every concentration, except *n* = 5 biological replicates for 16 μM EF-1), or after stimulation with NA-1 (dashed line with triangle; *n* = 3 biological replicates for every concentration). **d** Effect of $H_2O_2$ on cardiomyocyte cell death (*n* = 9 biological replicates for Vehicle group and *n* = 14 biological replicates for $H_2O_2$ group). **e** *Erbb4* knockdown efficiency of siRNA against *Erbb4* compared to control (i.e. scrambled) siRNAs in iAMs (*n* = 10 biological replicates for *SiScr* group and *n* = 13 biological replicates for *SiERBB4* group). **f** Effect of $H_2O_2$ on total cell death of iAMs in the presence of NA-1 (dashed line with triangle; *n* = 4 for every concentration, except *n* = 3 biological replicates for 1 μM NA-1), or in the presence of EF-1 (4–32 μM) after transfection with scrambled (full line with full circle; *n* = 3 biological replicates for every concentration) or *Erbb4*-specific siRNAs (dashed line with square; *n* = 3

biological replicates for every concentration). **g** CSA of iAMs after AngII exposure in the presence or absence of EF-1 (4–32 μM; *n* = 27 cell areas for Vehicle group, 38 for AngII + 8 μM EF-1 group and 40 for AngII group, AngII + 4 μM EF-1 group, AngII + 16 μM EF-1 group and AngII + 32 μM EF-1 group). Scale bar = 100 μm. **h** Effect of $H_2O_2$ on total cell death of iAMs in the presence or absence of the PI3K/AKT-pathway inhibitor LY294002, with and without EF-1 (32 μM, *n* = 3 biological replicates for Vehicle and $H_2O_2$/Vehicle group, *n* = 4 biological replicates for $H_2O_2$/EF-1 group and all groups with LY294002). **i** CSA of iAMs after AngII exposure in the presence or absence of the ERK-pathway inhibitor PD98059, with and without EF-1 (32 μM, *n* = 20 cell areas for all groups). All data are represented as mean ± SD, two-tailed unpaired t-test or one-way ANOVA with Dunnett's multiple comparisons test against respective concentration for *siScr*. *P*-values shown in (**c**) and (**f**) are between EF-1 + *siScr* and EF-1 + *siERRB4*. Source data are provided as a Source Data file. AngII angiotensin II, ANOVA analysis of variance, COL3A1 collagen type 3 alfa 1, CSA cross sectional area, FC fold change, $H_2O_2$ hydrogen peroxide, HCF human cardiac fibroblast, iAM conditionally immortalized rat atrial myocyte, M molar concentration, NA non-active compound containing the pharmacophore, SD standard deviation, *siERBB4* silencing RNA against *ERBB4*, *siScr* silencing RNA against scrambled control, TGF-β1 transforming growth factor β1, Veh vehicle.

pockets are potentially located at the domains involved in receptor dimerization, for instance domain II.

Both NRG1 and EF-1 induce ERBB4 receptor homodimerization, although the potency and efficacy of EF-1 is lower than those of NRG1. Remarkably, EF-1 also induced ERBB2-ERBB4 heterodimerization, with the same efficacy as NRG1. EF-1 induces phosphorylation of key proteins in 2 canonical pathways activated by NRG1: ERK1/2 and Akt. ERK1/2 phosphorylation induced by EF-1 is transient with peak levels at 15 min, which is in line with published data and our data on NRG1-induced ERK1/2 phosphorylation in neonatal rat ventricular myocytes[52]. Akt phosphorylation induced by EF-1, however, is much slower compared to NRG1-induced Akt phosphorylation, peaking at 2 h instead of 15 min[10]. Differences in potency, efficacy, signaling kinetics and the ability to induce ERBB dimerization pairs between EF-1 and NRG1 could at least partially explain the differences observed in Akt and ERK1/2 phosphorylation and in gene expression after the treatment of iAMs and HCFs with these compounds. The number of DEGs after EF-1 treatment of iAMs was significantly lower than after stimulation of the cells with the more potent NRG1 unlike the number of DEGs in HCFs, which were significantly higher after EF-1 treatment than after NRG1 stimulation. Finally, although we tested the effect of EF-1 in a number of contexts that required ERBB4 receptors including in cells transfected with ERBB4 targeting siRNA, and in *Erbb4*-null mice, and examined effects of non-active compounds that shared the same pharmacophore, off-target effects cannot completely be excluded both in vitro and in vivo. Additionally, ERBB4 signaling is complex, and even partial ERBB4 in vitro knockdown and in vivo knock-out experiments may significantly impact downstream signaling pathways.

It is know that the NRG1/ERBB signaling system has distinct effects in male vs. female mice[53]. For that reason, we investigated both male and female sexes in our in vivo experiments and we observed both shared and sex-specific responses across the three models of cardiac injury. In the angiotensin-II-induced myocardial fibrosis model, both male and female mice exhibited a significant reduction in total and perivascular fibrosis, suggesting that the intervention attenuates fibrotic remodeling in both sexes. However, in the doxorubicin-induced cardiotoxicity and myocardial infarction models, we observed a pronounced sex-specific effect, with only female mice showing significant improvements in cardiotoxicity and post-infarction heart function, respectively. These findings highlight potential sex differences in the response to cardiac injury, particularly in cardioprotection and recovery mechanisms, which may be influenced by hormonal or molecular factors unique to females. This underscores the importance

of considering sex as a biological variable in preclinical studies and warrants further investigation.

In summary, we showed that small molecules can act as ERBB4 agonists inducing ERBB4 dimerization and triggering ERBB4-mediated biological effects in fibroblasts and cardiomyocytes. We also showed in vivo evidence that these small molecules could be a novel therapeutic strategy for treatment of CHF. As ERBB4 is also important in other diseases like fibrotic, inflammatory, and neurological disorders[11,54–58], small-molecule ERBB4 agonists could also be of therapeutic relevance in other diseases.

## Methods

### Study approval

All animal experiments were approved by the Ethical Committee of the University of Antwerp (approval number ECD 2020-08) and conformed to the Guide for the Care and Use of Laboratory Animals, 8th edition published by the US National Institutes of Health in 2011, and to the European Communities Council Directive 2010/63/EU for the protection of animals used for experimental purposes. All animals were fed on a standard chow, were provided with water *at libitum*, and were housed at a constant temperature of 22 °C and humidity of 50% in a 12 h controlled light/dark cycle. Throughout the experimental period, mice were closely monitored for any signs of distress or adverse effects.

### Cells

PathHunter U2OS ERBB4/ERBB4 (osteosarcoma cells; Eurofins, 93-0961C3), U2OS ERBB2/ERBB4 (osteosarcoma cells; Eurofins, 493-0960C3), ERBB2/ERBB3 (osteosarcoma cells; Eurofins, 93-1042C3) and ERBB1/ERBB1 (osteosarcoma cells; Eurofins, 93-0989C3) dimerization cell lines were cultured according to the manufacturer's instructions. Briefly, cells were cultured in Cell Culture Reagent 103 (Eurofins, 92-3103 G) supplemented with 250 μg/mL Hygromycin B (Eurofins, 92-0029) and 500 μg/mL G418 (Eurofins, 92-0030), and were maintained at 37 °C in humidified atmosphere of 5% $CO_2$. HCFs (Innoprot, P10454) were cultured in fibroblast medium (Innoprot, P60108) supplemented with 10% (v/v) fetal bovine serum (FBS, Innoprot), 10% (v/v) fibroblast growth supplement (Innoprot), and 1% (v/v) penicillin/streptomycin solution (Innoprot). iAMs[33] (kindly provided by LUMC) were cultured in Advanced DMEM F-12 (Thermo Fisher Scientific, 12634028) supplemented with 2% (v/v) heat-inactivated FBS (Thermo Fisher Scientific, 10270106), 1% (v/v) penicillin/streptomycin (10,000 units/mL and 10 mg/mL respectively, Thermo Fisher Scientific, 15140122), 1× Gluta-MAX (Thermo Fisher Scientific, 35050061), and 100 ng/mL

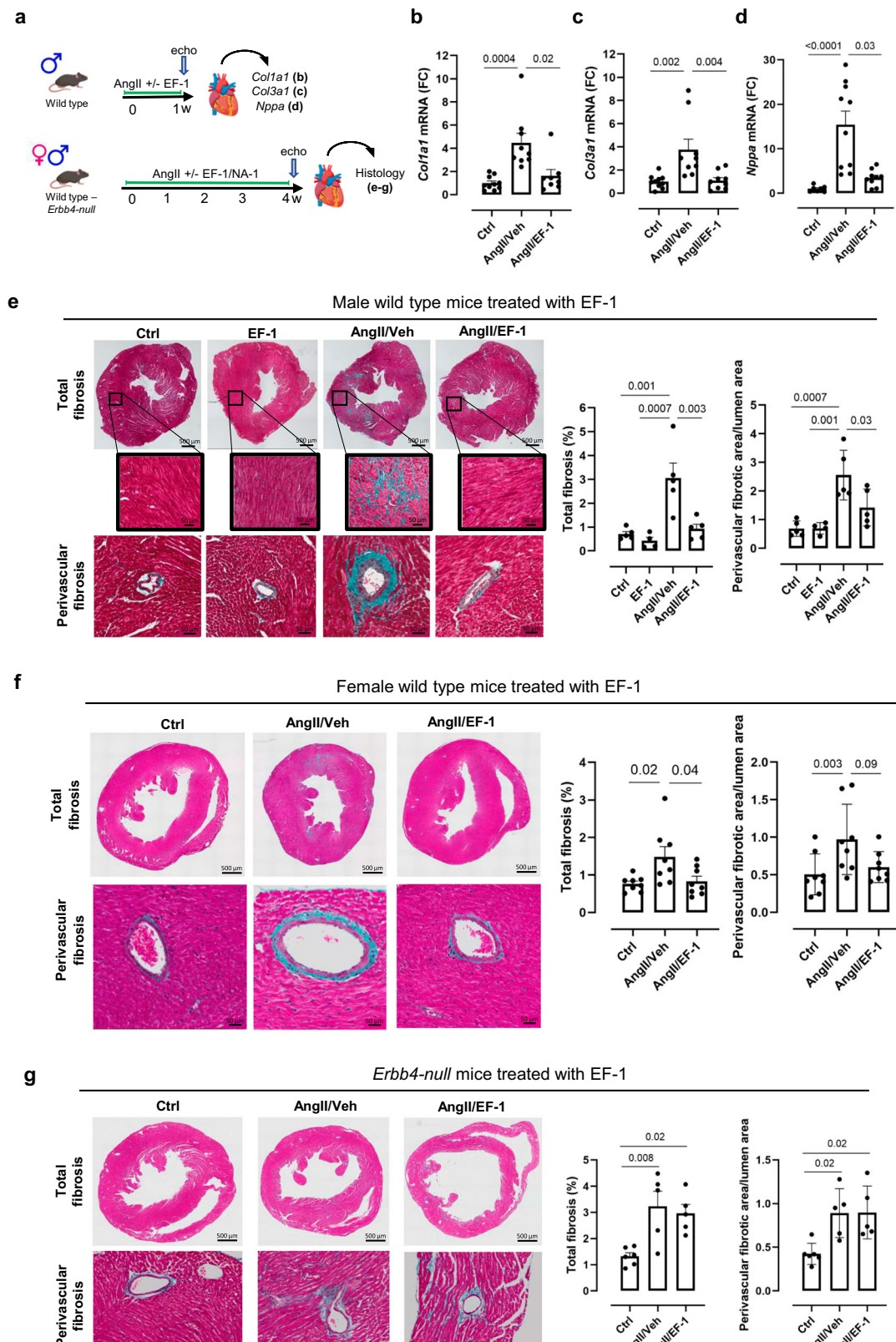

doxycycline (Tocris, 4090) for proliferation. To induce differentiation, iAMs were transferred to medium without doxycycline.

## HTS and chemoinformatics

An HTS of 10,240 compounds (Pharmacological Diversity Set, Enamine) was performed using the PathHunter U2OS ERBB4/ERBB4

dimerization cell line. U2OS ERBB4/ERBB4 dimerization cells were seeded at a density of $5 \times 10^3$ cells/well in 50 μL Cell Plating 0 Reagent (Eurofins, 93-0563R0A) in white 384-well plates (Greiner Bio-One, 781080). Cells were treated with compound (10 μM), NRG1 (positive ctrl, 1 μM; Eurofins, 92-1031), or phosphate-buffered saline (negative ctrl, PBS; Thermo Fisher Scientific, 14040133). All wells contained

**Fig. 5 | EF-1 prevents cardiac fibrosis in vivo. a** Design of the in vivo experiments. Created in BioRender. Cools, J. (2024) https://BioRender.com/y82u338. Hearts were collected after 1 week for mRNA analysis and after 4 weeks for histological analysis. **b**–**d** RT-qPCR was performed on fibrosis and cardiac stress markers; normalized FC compared to Ctrl: **b** *Col1a1* (*n* = 10 biological replicates in ctrl group, *n* = 9 biological replicates in AngII/Veh group and *n* = 8 biological replicates in AngII/EF-1 group), **c** *Col3a1* (*n* = 10 biological replicates in ctrl group and *n* = 9 biological replicates in AngII/Veh and AngII/EF-1 group) and **d** *Nppa* (*n* = 7 biological replicates in ctrl group, *n* = 10 biological replicates in AngII/Veh group and *n* = 9 biological replicates in AngII/EF-1 group). **e** Representative images of Masson's trichrome staining of AngII-induced myocardial fibrosis following treatment with EF-1 or Veh and corresponding bar graphs showing the quantitation for total, interstitial and perivascular fibrosis in male wild type mice (*n* = 4 mice in EF-1 group and *n* = 5 in all other groups). **f** Representative images of Masson's trichrome staining of AngII-induced myocardial fibrosis following treatment with EF-1 or Veh and corresponding bar graphs showing the quantitation for total and perivascular fibrosis in female wild type mice (*n* = 8 mice in all groups). **g** Representative images of Masson's trichrome staining of AngII-induced myocardial fibrosis in male and female *Erbb4-null* mice, treated with EF-1 or Veh and corresponding graphs showing the quantitation for total and perivascular fibrosis (*n* = 6 mice in Ctrl group and *n* = 5 in all other groups). Scale bar = 500 µm for total fibrosis, 50 µm for interstitial and perivascular fibrosis. All data are represented as mean ± SD, one-way ANOVA with Tukey's multiple comparisons test. Source data are provided as a Source Data file. AngII angiotensin II, ANOVA analysis of variance, Col1a1 collagen type 1 alpha 1, Col3a1 collagen type 3 alpha 1, Ctrl control, echo echocardiography, FC fold change, NA non-activating pharmacophore-containing compound, Nppa atrial natriuretic peptide, RT-qPCR reverse transcription-quantitative polymerase chain reaction, SD standard deviation, Veh vehicle.

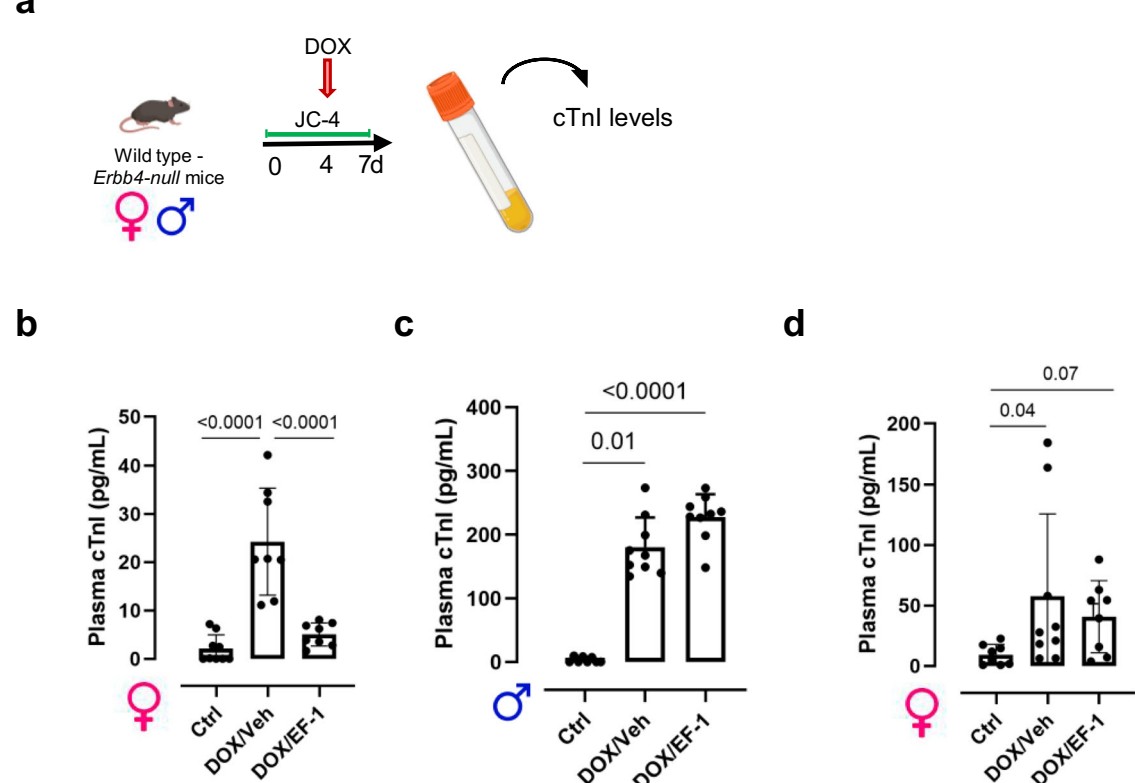

**Fig. 6 | EF-1 prevented cardiomyocyte cell death in vivo in female mice. a** Design of the in vivo DOX experiment. Created in BioRender. Cools, J. (2024) https://BioRender.com/y82u338. Graphs showing the cTnI levels measured in plasma samples of untreated control mice and of mice treated for 7 days with EF-1 or Veh and given a single IP injection of 20 mg/kg DOX on day 4 of treatment for **b** female wild type mice (*n* = 9 mice in Ctrl group, *n* = 8 mice in all other groups), **c** male wild type mice (*n* = 8 mice in Ctrl group, *n* = 9 mice in all other groups) and **d** *Erbb4*-null mice (*n* = 9 mice in DOX/Veh group, *n* = 8 mice in all other groups). All data are represented as mean ± SD, one-way ANOVA test with Tukey's correction for multiple testing. Source data are provided as a Source Data file. ANOVA analysis of variance, cTnI cardiac troponin I, Ctrl control, DOX doxorubicin, IP intraperitoneal, SD standard deviation, Veh vehicle.

dimethyl sulfoxide (DMSO; Merck Life Science, D2438) at a final concentration of 1%. The cells were subsequently incubated for 6 h at 37 °C in a humidified atmosphere of 5% $CO_2$. Then, 25 µL of PathHunter Flash Detection Reagent (Eurofins, 93-0247) was added to each well, and cells were incubated at room temperature (RT) in the dark for 1 h. Subsequently, luminescence was measured using the EnVision plate reader (Revvity). Spotfire (TIBCO) was used for data analysis and visualization and Collaborative Drug Discovery Vault for compound registry and data management. Data from the primary screening was analyzed via the HTS-Corrector software[59]. In HTS-Corrector, intra-plate normalization (via median polish) was performed to correct for row, column, or edge effects. The analysis output was the normalized

values for each compound. Subsequently, normalized values were used to perform inter-plate normalization, which generated B-scores as final analysis output. The top 80 compounds having the highest B-score were selected and re-tested in a confirmation screen under the same assay conditions as the primary screen. In the confirmation screen, the threshold for hit selection was defined as the average signal of the negative control plus three times the standard deviation (SD) of the negative control.

To validate and cluster hit compounds, chemoinformatic methods were used to identify the MCSs between the biological active compounds. To estimate the relative significance of each MCS pattern for biological activity, the enrichment of each MCS

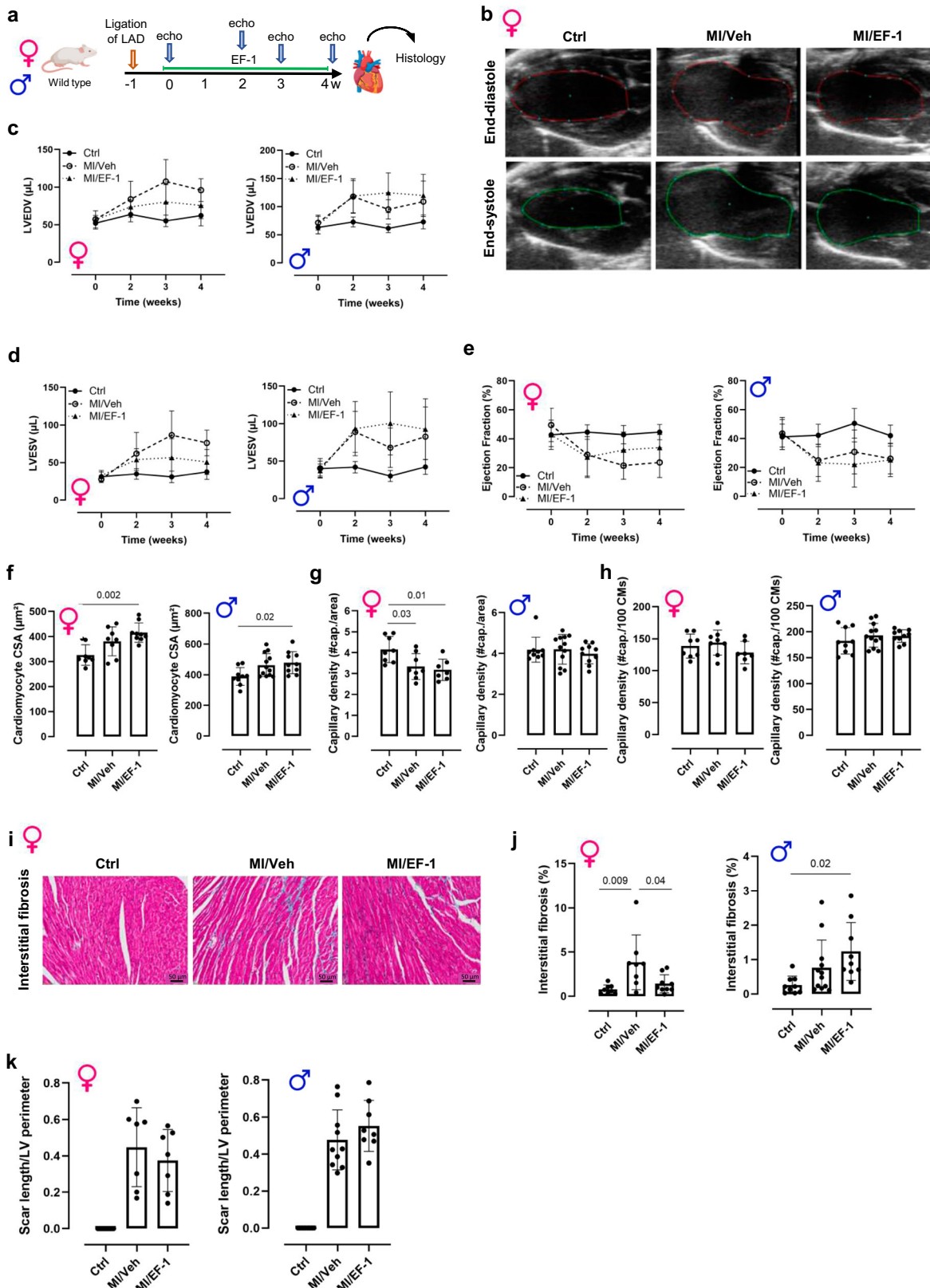

pattern was calculated by comparing the occurrence of the respective MCS pattern in both the hitlist and reference set (which was the entire Enamine HTS collection consisting of 1,773,567 compounds). MCS patterns with the highest enrichment were then used to select compounds from the Enamine library for additional screening that contain the respective MCS from the Enamine library, resulting in an additional 111 compounds for follow-up screening. In parallel, we developed a random-forest machine-learning model using Morgan fingerprints of active and inactive molecules of previous screenings.

**Fig. 7 | EF-1 decreases ventricular dilation after MI in female mice. a** Design of the in vivo MI experiment. Created in BioRender. Cools, J. (2024) https://BioRender.com/y82u338. **b** Representative echocardiographic images in end-diastole and end-systole of sham-operated control female mice and of mice that underwent MI for 4 weeks with EF-1 or Veh. Graphs showing (**c**) LVEDV, (**d**) LVESV and (**e**) EF in the different experimental groups during the entire experiment in both female and male mice. Ctrl = full line with full circle; MI/Veh = dashed line with open circle; MI/EF-1 = dashed line with triangle. **f** CSA of cardiomyocytes analyzed with an isolectin-B4/WGA/DAPI staining in female and male mice. Vascularization analysis of capillaries with an isolectin-B4/WGA/DAPI staining, showing the number of capillaries in female and male mice (**g**) per area of cardiomyocytes and (**h**) per 100 cardiomyocytes. **i** Representative images of Masson's trichrome staining of interstitial fibrosis in the remote zone in female mice. **j** Corresponding graph quantifying the fibrotic area in the different experimental groups in female mice and the graph of the quantification of the fibrotic area in male mice. **k** Scar size analysis of the myocardial infarction area in female and male mice. Scale bar = 50 μM for interstitial firosis. All data are represented as mean (biological replicates) ± SD, $n = 8$ female mice in Ctrl and MI/Veh group, $n = 9$ female mice in MI/EF-1 group; $n = 10$ male mice in Ctrl group and MI/EF-1 group on T2 and T4, $n = 5$ male mice in Ctrl group and MI/EF-1 group on T0 and T3, $n = 12$ male mice in MI/Veh group on T2 and T4, $n = 6$ male mice in MI/Veh group on T0 and T3, one-way ANOVA test with Tukey's correction for multiple testing. Source data are provided as a Source Data file. ANOVA analysis of variance, CSA cross-sectional area, Ctrl control, DAPI 4′,6-Diamidino-2-phenylindole dihydrochloride, EF ejection fraction, IP intraperitoneal, LVEDV left-ventricular end-diastolic volume, LVESD left-ventricular end-systolic volume, MI myocardial infarction, SD standard deviation, Veh vehicle, WGA wheat germ agglutinin.

## PathHunter dimerization assay in 96-well format
U2OS ERBB4/ERBB4 dimerization cells were seeded at a density of $5 \times 10^4$ cells/well in 100 μL Cell Plating 0 Reagent, in white 96-well plates (PerkinElmer, 6005680) and incubated for 24 h at 37 °C, 5% $CO_2$. Cells were treated for 6 h with different concentrations of the 8 hit compounds (Enamine, EF-1 – EF-8) or compound NA-1, NRG1 (Peprotech, 100-03), or PBS. Next, 110 μL PathHunter Flash Detection Reagent was added to each well and incubated at RT for 1 h after which the luminescence signal was measured using the Luminoskan Ascent (Thermo Fisher Scientific). Dose-response curves of the compounds were performed in *2-fold* (0.0625–32 μM). The same experimental set-up was used for the U2OS ERBB2/ERBB4, U2OS ERBB2/ERBB3 and U2OS ERBB1/ERBB1 dimerization cell lines. For co-administration of NRG1 and EF-1, cells were pretreated for 10 min with 5 μL of EF-1 (1, 10, or 32 μM) before adding 5 μL of different NRG1 concentrations and incubated for 6 h at 37 °C, 5% $CO_2$. All wells contained DMSO at a final concentration of 0.9%.

## Fluorescence-based competition binding assay
NRG1 was dissolved in PBS at a concentration of 1 mg/mL and labeled with Alexa Fluor 488 using the Alexa Fluor 488 microscale protein labeling kit (Thermo Fisher Scientific, A30006). A Bio-gel P-4 (Bio-Rad, 1504124) fine resin suspended in PBS and a dye:protein molar ratio of 5 was used to purify the labeled NRG1 according to the manufacturer's instructions. Fluorescent labeling was evaluated by performing a dose-response experiment using F-NRG1 (1–1000 nM) on the U2OS ERBB4/ERBB4 dimerization cell line. Next, the same cell line was used to perform a competition binding assay with F-NRG1 using flow cytometry. Cells were plated in transparent U-bottom 96-well plates (Greiner Bio-One, M9436) at $5 \times 10^5$ cells/mL in 50 μL ice-cold buffer (PBS with 0.1% bovine serum albumin (BSA; Sigma-Aldrich, A7906) and 0.05% sodium azide (Sigma-Aldrich, S2002)). Next, cells were centrifuged at $17,968 \times g$ for 4 min at RT and the supernatants were discarded. The cell pellets were washed with 50 μL ice-cold buffer by gently pipetting up and down. Centrifugation and washing were repeated, and the supernatants were discarded. For the competition assay between NRG1 and F-NRG1, cells were treated with NRG1 (1–100 nM) and F-NRG1 (30 nM). For the competition assay between EF-1 and F-NRG, cells were treated with EF-1 (0.1–100 μM) and F-NRG1 (30 nM). All wells contained DMSO at a final concentration of 0.9%. After gentle mixing of each sample, the plate was incubated for 1 h at 4 °C on a microplate shaker. Next, the plate was centrifuged at $17,968 \times g$ for 4 min at RT and the supernatants were discarded. The cells were then washed with 100 μL ice-cold buffer, pelleted by centrifugation and washed again. The supernatants were discarded and 100 μL ice-cold buffer was added to the cell pellets and pipetted up and down to generate single-cell suspensions. These suspensions were transferred to 5-mL polystyrene round-bottom tubes, kept on ice, and exposed to minimum light until flow cytometric analysis (BD Accuri C6, BD Biosciences). Unstained cells were used to set the parameters of the flow cytometer.

## RNA sequencing
iAMs were seeded at a density of $6 \times 10^6$ cells/mL in differentiation medium in 75-$cm^2$ cell culture flasks (Greiner Bio-One, 658175) and incubated for 9 days (d0 = day of seeding). HCF were seeded at a density of $3 \times 10^6$ cells/mL in fibroblast medium in 75-$cm^2$ cell culture flasks. Then, cells were incubated for 16 h at 37 °C in the presence of EF-1 (32 μM), PBS, or NRG1 (0.1 μM). All wells contained DMSO at a final concentration of 0.9%. Next, cells were lysed using 350 μL buffer RLT (Qiagen) supplemented with β-mercaptoethanol (100:1; Merck, 444203). Total RNA was isolated using the RNeasy Micro Kit (Qiagen, 74104) according to the manufacturer's protocol, with an extra step of DNase digestion. The concentration and quality of RNA were determined using a Qubit fluorometer (Thermo Fisher Scientific) and 2100 Agilent BioAnalyzer (Agilent Technologies), respectively. Samples with an RNA integrity number >7 were used for library preparation. Sequencing libraries were prepared by Genewiz/Azenta (Leipzig) on cDNA prepared from polyadenylated mRNA. Libraries were sequenced using a NovaSeq 6000 (Illumina) with a read length of $2 \times 150$ bp.

Gene expression was quantified at the transcript level using Salmon (v1.10.0)[60], with the validatMappings and -gcBias parameters switched on, to the Rnor_6.0 or GRCh38 transcriptome. Transcript level counts were aggregated to gene level using the import in the tximport package (v1.26.1)[61], setting countsFromAbundance to 'lengthScaledTPM' in R (v4.1.1). DESeq2 R package (v1.38.3)[62] was used for differential gene expression analysis between different conditions. The batch variability of different sequencing runs was accounted for by defining "batch" as a covariate in the linear model to analyze differential gene expression. Differential gene expression heat-maps were generated by using the pheatmap R package (v1.0.12), and volcano plots by EnhancedVolcano (v1.10.0). The overlapping genes between different conditions were obtained by VennDiagram (v1.7.3)[63]. Functional enrichment of DEGs was determined using a hypergeometric test against the Gene Ontology database by using the ClueGO (v2.5.7)[64] module of Cytoscape (v3.9.1)[65] with Benjamini–Hochberg adjusted (FDR) $P < 0.05$. GSEA Preranked method was performed to identify the Hallmark pathways[66].

The RNA sequencing data generated has been deposited in NCBI's Gene Expression Omnibus (https://www.ncbi.nlm.nih.gov/geo) and is accessible through GEO Series accession numbers GSE256024 for iAM data and GSE261219 for HCF data.

## SiRNA transfection experiment
SiRNA transfection was carried out according to the manufacturer's instructions, using 1.25 μL DharmaFECT 1 transfection reagent (Horizon Discovery, T-2001) and 2.5 μL of 10 μM *ERBB4* siRNAs or non-targeting siRNAs (ON-TARGETplus Human *ERBB4* or Non-Targeting Pool, Horizon Discovery, D-00180/T-2001) per well. After the assay,

cells transfected with and without *ERBB4* siRNAs, to determine knockdown efficiency, were lysed using RA1 lysis buffer (Macherey-Nagel, 740955.250) supplemented with β-mercaptoethanol (100:1), scraped with a cell scraper and collected in an Eppendorf tube (Greiner Bio-One, 616201) before RNA isolation for reverse transcription-quantitative polymerase chain reaction (RT-qPCR).

## In vitro collagen expression

HCFs were seeded at a density of $1.5 \times 10^5$ cells/well in doxycycline-free medium in 12-well plates (Greiner Bio-One, 665180) and incubated overnight. Next, siRNA transfection was carried out as described above. Cells were incubated for 22 h at 37 °C and 5% $CO_2$. Next, medium was changed to fresh differentiation medium (without doxycycline), and cells were stimulated for 24 h with either PBS, EF-1 (4−32 μM) or NA-1 (4−32 μM) together with TGF-β1 (10 ng/mL; Peprotech, 100-21). All wells had a final concentration of 0.9% DMSO. Compounds were pre-incubated for 10 min before addition of TGF-β1. Next, cells were lysed using 100 μL RA1 lysis buffer supplemented with β-mercaptoethanol (100:1), scraped with a cell scraper, and collected in an Eppendorf tube before RNA isolation for RT-qPCR.

## H₂O₂−induced cell death assay

iAMs were seeded at a density of $2.7 \times 10^5$ cells/well in 48-well plates (Greiner Bio-One, 677180) and differentiated over 9 days in medium without doxycycline (d0 = day of seeding). Next, siRNA transfection was carried out as described above. After incubation for 24 h at 37 °C and 5% $CO_2$, the culture medium was refreshed. Next, the cells were pretreated for 10 min with either EF-1 (4−32 μM), PBS, or NA-1 (4−32 μM). Subsequently, $H_2O_2$ (100 μM; Merck, H1009) was added and the cells were incubated for 4 h at 37 °C. All wells contained DMSO at a final concentration of 0.9%. Next, 150 μL of the culture medium in each well was transferred to a 96-well plate and mixed with 100 μL Toxilight AK detection reagent (Lonza, LT07-217). After incubation at RT for 5 min, luminescence was measured using the Luminoskan Ascent. A 100% lysis control (Lonza, LT27-239) was used to determine the percentage of total dead cells.

To confirm that EF-1 decreased $H_2O_2$-induced cell death, we repeated the experiment with two different assays (WST1 and TUNEL).

Following the transfer of 150 μL of culture medium for the Toxi-Light assay, 7 μL of WST-1 reagent (Merck, 5015944001) was added directly to the remaining cell culture medium in each well. The cells were incubated at 37 °C for an additional 4 h. After incubation, the absorbance of the samples was measured at 450 nm (690 nm was used as reference wavelength and subtracted) using a microplate reader (Epoch).

To assess apoptosis, a TUNEL assay was performed using the In Situ Cell Death Detection Kit, Fluorescein (Merck, 11684795910) according to the manufacturer's instructions. After the 4-h incubation, the cells were fixed with 4% paraformaldehyde for 15 min at room temperature, followed by permeabilization with 0.1% Triton X-100 in 0.1% sodium citrate for 2 min on ice. Cells were then incubated with the TUNEL reaction mixture in a humidified chamber at 37 °C for 1 h in the dark. After washing with PBS, the samples were analyzed using fluorescence microscopy to detect apoptotic cells.

## Cardiomyocyte hypertrophy assay

iAMs were seeded at a density of $10^4$ cells/well in differentiation medium in 24-well plates (Greiner Bio-One, 662160) and differentiated over 9 days (d0 = day of seeding). Next, the cells were pretreated for 1 h with either EF-1 (4−32 μM) or PBS after which AngII (100 nM; Merck, A9525) was added and the cells were incubated for 24 h at 37 °C. All wells contained DMSO at a final concentration of 0.9%. To determine CSA, cells were fixed with 4% paraformaldehyde (PFA, Thermo Fisher Scientific, 043368.9 M) for 30 min at 4 °C. Cells were washed thrice with PBS and permeabilized by incubation with 0.1% Triton X-100

(Merck, 10789704001) for 10 min at RT. iAMs were stained with Alexa Fluor 568 phalloidin (Thermo Fisher Scientific, A12380) in 1% BSA-PBS solution for 1 h at RT. After washing 3 times with PBS, DAPI (Merck, D9542) was added. Images were obtained by fluorescence microscopy (Celena S).

CSA was quantified using an automated algorithm, custom-made in Python[67]. The mode $m_D$ (maximum of the histogram) of DAPI intensities (excluding 0) was determined and binary segmentation of nuclei was created with threshold $m_D + SD_D$ (with $SD_D$ the SD of DAPI intensities). Distinct objects with an area less than 500 pixels and nuclei where the average overlapping iAM signal was less than $md_A + 0.5\,SD_A$ (with $md_A$ and $SD_A$ the median and SD of the Alexa Fluor 568 intensity, respectively) were removed. Binary segmentation of Alexa Fluor 568 signal was created using a threshold of $m_A + 0.5\,SD_A$ (with $m_A$ the mode of Alexa Fluor 568 intensities excluding 0), followed by one pass of binary erosion and two passes of binary dilation, each with a 1-connected neighborhood. Any pixels with intensities less than $m_A\ 0.5\,SD_A$ were removed from the segmentation. Next, a gradient image was created from the Alexa Fluor 568 phalloidin channel. A gray erosion on the original Alexa Fluor 568 intensities (structuring element $3 \times 3$ pixels) was followed by a Gaussian blur ($\sigma = 3$ pixels) and a 2D Scharr filter as implemented by 'scikit-image'. Intensities of the resulting gradient image were scaled to the full 8-bit range, before another gray dilation was performed with a $5 \times 5$ pixel structuring element to smooth the gradient. The gradient was used for watershed segmentation of the individual cells with segmented nuclei as seeds. The implementation provided by 'scikit-image' was used with a *compactness* parameter of 0.1. All objects extending to the image borders were removed. Holes within each remaining individual object were filled, and 3 iterations of binary erosion followed by 3 iterations of binary dilation removed small protrusions. For each cell, the CSA was reported.

## In vitro pathway inhibitors

LY294002 (MedChem Express, HY-10108) was used to inhibit the PI3K/AKT pathway. iAMs were treated with EF-1 (32 μM) or vehicle, with or without 10 min pretreatment with LY294002 (20 μM) and cell death was induced with $H_2O_2$.

PD98059 (MedChem Express, HY-12028) was used to inhibit the ERK1/2 pathway. iAMs were treated with EF-1 (32 μM) or vehicle, with or without 120 min pretreatment with PD98059 (50 μM) and iAM hypertrophy was induced with AngII.

## EdU proliferation assay

EdU proliferation assay was performed using the Click-iT™ EdU Cell Proliferation Kit for Imaging (ThermoFisher Scientific, C10337) following the protocol. Proliferating iAMs were seeded at a density of $10^5$ cells/well in proliferating medium in 24-well plates and incubated overnight. The cells were stimulated with either vehicle, EF-1 (32 μM) or NRG1 (0.1 μM) for 24 h at 37 °C. All wells contained DMSO at a final concentration of 0.9%. Afterwards, the cell culture media were then replaced with 10 μM EdU in proliferating medium and incubated for 24 h at 37 °C. Next, iAMs were fixed with 4% PFA for 30 min at 4 °C. Cells were washed thrice with PBS, permeabilized by incubation with 0.1% Triton X-100 for 10 min at RT and counterstained with DAPI. Images were obtained by fluorescence microscopy (Celena S).

## In vitro assays to assess compound stability in plasma and in liver microsomes

A 5-μL aliquot of compound solution (10 mM in DMSO) was added to 995 μL of Non-Swiss Albino Mouse Plasma (Innovative Research, IMSNSAPLAK2E10mL) or Pooled Normal Human plasma (Innovative Research, IPLAK2E10ML) in sodium citrate to obtain a final concentration of 50 μM compound in plasma. The mixture was gently shaken for 6 h at 37 °C. Aliquots of 100 μL were taken at various time

points (0, 0.5, 1, 2, and 6 h), and diluted with 400 μL of ice-cold acetonitrile (Sigma-Aldrich, AX0156). The resulting suspensions were centrifuged at 17,968 × g for 5 min. Subsequently, 50 μL of the supernatant was diluted with 950 μL of ice-cold acetonitrile and analyzed by liquid chromatography with tandem mass spectrometry (LC-MS/MS; Waters Acquity H-class UPLC system with a Bruker Daltonics Esquire 3000 plus ion trap mass spectrometer and an Agilent 1100 Series LC system). Two biological replicates were analyzed in triplicate (three technical replicates) and plotted against a standard curve (compound at 31–1000 nM in plasma and diluted in ice-cold acetonitrile as described above).

A mixture of 713 μL milliQ water (Merck, C85358), 200 μL 0.5 M phosphate buffer (pH 7.4, Becton Dickinson, TBS5034), 50 μL NADPH regenerating system solution A (Becton Dickinson), 10 μL NADPH regenerating system solution B (Becton Dickinson) and 2 μL compound (5 mM in DMSO) was prepared and heated for 5 min at 37 °C. A volume of 25 μL human and mouse liver microsomes (0.5 mg protein/mL, Corning Life Sciences, 452117 and 452220, respectively) was added to the mixture and 20 μL samples were withdrawn at 0, 0.25, 0.5, 1, 2, 4, 6 and 24 h. Next, 80 μL of ice-cold acetonitrile was added to the samples. After a 10-min incubation period on ice, the mixtures were centrifuged at 15,493 × g for 5 min at 4 °C. Finally, 75 μL of an acetonitrile/water (10/90) mixture was added to 25 μL of supernatant and the resulting samples (three biological replicates) were analyzed in triplicate (three technical recplicates) by LC-MS/MS.

A control sample was measured for every sample measurement at time point 0 and a blank sample was always measured for every sample measurement before a new time point. Details about sample preparation, the implementation of the protocols and the raw data files can be found in the Supplementary Data 1.

### Mouse model of AngII-induced myocardial fibrosis
Thirteen-week-old C57BL/6N (Charles River, 027) female and male mice were randomized to the ctrl group (n = 5 male mice, n = 8 female mice) or to the groups treated with EF-1 (EF-1 group, n = 4 male mice), AngII plus vehicle (AngII/Veh group; n = 5 male mice, n = 8 female mice), or AngII plus EF-1 (AngII/EF-1 group; n = 5 male mice, n = 8 female mice). AngII (1000 ng/kg/min in PBS), Veh (DMSO/propylene glycol/50:50), and EF-1 (2 mg/kg/day in Veh) were administered for 4 weeks using subcutaneously implanted micro-osmotic pumps (Alzet, model 1004). Four weeks after implantation, cardiac ultrasound was performed, mice were euthanized, and hearts were collected. A similar set-up was used for the 1-week study (Alzet, model 1007D) in male mice (n = 9-10 mice in every group). In some experiments, EF-1 was replaced by NA-1 (2 mg/kg/day in Veh; n = 5 male mice in ctrl and AngII/Veh group, n = 4 male mice in AngII/EF-1 group).

### Mouse model of acute high-dose DOX-induced cardiotoxicity
Twelve-week-old C57BL/6 N male and female mice were randomized to the ctrl group (n = 9 female mice, n = 8 male mice), or to the groups treated with DOX plus Veh (DOX/Veh group; n = 8 female mice, n = 9 male mice), or DOX plus EF-1 (DOX/EF-1 group; n = 8 female mice, n = 9 male mice). EF-1 (2 mg/kg in Veh) or Veh were administered for 1 week using subcutaneous micro-osmotic pumps and started on day 1. DOX (20 mg/kg; Pfizer, 4222) was administered intraperitoneally once on day 4 to induce cardiotoxicity. On day 7, mice were euthanized, serum samples were collected via the retrobulbar sinus and hearts were excised.

### Mouse model of MI
Twelve-week-old female and male Balb/cJ (Charles River, 000651) mice were randomized into either the ctrl or MI group. Mice in the MI group underwent surgical ligation of the LAD, while mice in the ctrl group underwent a sham procedure (n = 8 female mice, n = 10 male mice). Briefly, all mice received an injection of 0.1 mg/kg buprenorphine

(Produlab Pharma) before induction of anesthesia with 8% sevoflurane (Zoetis). Anesthesia was maintained with 4.5% sevoflurane and mice were intubated and ventilated. An incision of 15 mm was made on the left side of the thorax and the thoracic cavity was opened at the third intercostal space using blunt forceps. The LAD was permanently ligated with a 8/0 polypropylene monofilament sutures (Ethicon, F1894) after which the thoracic cavity and skin were closed with 6/0 polypropylene monofilament sutures (Ethicon, F1841). and 5/0 polyamide 6 sutures (Ethilon, F2412H), respectively. Mice received another dose of 0.1 mg/kg buprenorphine 6–8 h after the initial dose. Two days after surgery, cardiac ultrasound was performed to exclude mice without successful MI (i.e. displaying hypokinesia in ≥2 out of 5 segments). Mice with successful MI were randomized into 2 groups: an MI/EF-1 group (n = 9 female mice, n = 10 male mice) and an MI/Veh group (n = 8 female mice, n = 12 male mice). Seven days after MI surgery, the mice were equipped with osmotic mini-pumps containing either EF-1 (2 mg/kg/day) or Veh, as described above. Cardiac ultrasound was performed weekly. After 4 weeks of treatment, mice were euthanized, and hearts were collected.

### Transgenic mouse models
Floxed Erbb4 mice (Erbb4$^{f/f}$; C57BL/6N; 129-Erbb4tm1Fej/Mmucd, MMRRC, #010439-UCD) were crossed with CAGGCre-ER$^{TM}$ mice (Jackson Laboratory, 004682) containing the Tg(CAG-cre/Esr1*)5Amc transgene that expresses Cre recombinase under the control of a chicken beta actin promoter/enhancer coupled to the human cytomegalovirus immediate-early gene enhancer, resulting in expression of tamoxifen-inducible Cre-ERT in most cell types[68]. To induce Cre recombinase-mediated deletion of Erbb4, eleven-week-old female and male iCAGGCre-ER$^{TM}$/Erbb4$^{f/f}$ mice were intraperitoneally injected with tamoxifen (Merck, T5648; 10 mg/kg; in corn oil) daily for 5 consecutive days. Mice were used in aforementioned studies two weeks after the start of the tamoxifen injections (for the AngII-experiment with EF-1, n = 3 male and 3 female ctrl mice, n = 2 male and 3 female mice in the AngII/Veh group, n = 3 male and 2 female mice in the AngII/-EF-1 group; for the DOX-experiment, n = 8 female ctrl mice, n = 9 female DOX/Veh mice, n = 8 female mice in the DOX/EF-1 group; for the survival analysis, we used 3 male and 3 female ctrl mice and 9 male and 6 female AngII/Veh mice). As a control experiment, eleven-week-old female and male iCAGGCre-ER$^{TM}$/Erbb4$^{f/f}$ mice were injected with corn oil without tamoxifen (n = 4 male and 4 female ctrl mice, n = 5 male and 5 female mice in the AngII/Veh group and n = 4 male and 4 female mice in the AngII/EF-1 group).

### Cardiac ultrasound
Echocardiography was performed using the Vevo F2-LAZRX (FUJIFILM VisualSonics) and UHF57x probe. Mice were anesthetized with 1.5% isoflurane (Alvira, BE-V512222). Parasternal long-axis B-mode images and M-mode images were obtained and analyzed using VevoLAB software (FUJIFILM VisualSonics, Version 5.7.1). For the AngII experiments, measurements were obtained through analysis of the short-axis M-mode. For the MI experiments, measurements were obtained through analysis of the parasternal long-axis B-mode, of which the area and end-volume of the left ventricular cavity in diastole and systole were determined by tracing the endocardial border. Acquisitions, measurements and analyses were performed blinded.

### Serum cTnI enzyme-linked immunoassay
Serum samples were centrifuged for 15 min (1000 × g) at 4 °C and analyzed using a mouse cardiac Troponin I Type 3 ELISA kit (Novus Biologicals, NBP3−00456), according to the manufacturer's instructions. Briefly, a 96-well microplate was coated with an anti-mouse TNNI3/cTnI primary antibody (Capture Antibody Solution). Standard solution or samples (100 μL) were added to the plate and incubated for 90 min at 37 °C. A biotinylated anti-mouse TNNI3/cTnI detection antibody was added, and the plate was incubated for 1 h at 37 °C

followed by 3 washes with Wash Buffer. An avidin–horseradish peroxidase conjugate (Detection Antibody Solution) was then added, and the plate was incubated once more for 30 min at 37 °C. After 3 washes with Wash Buffer, Substrate Reagent was added. Following incubation for 15 min at 37 °C, the enzymatic reaction was terminated by the addition of Stop Solution. Sample optical densities at 450 nm were converted to tissue concentrations of mouse TNNI3/cTnI using a calibration curve.

### Western blot analysis

iAMs were seeded at a density of $10^6$ cells/well in a transparent 6-well plate (Greiner Bio-One, 657160) and treated with either NRG1 (10 nM), EF-1 (32 μM) or PBS for 0, 15, 60, 120, or 240 min; all wells contained DMSO at a final concentration of 0.9%. Hearts of the aforementioned transgenic mice were removed immediately after the animals had been killed. Cells and heart tissue were collected in RIPA lysis buffer (Thermo Fisher Scientific, 89900) supplemented with protease inhibitors (Merck, 11836153001) and phosphatase inhibitors (Merck, 4906845001) and the lysates were centrifuged at $14,000 \times g$ for 10 min. The supernatants were collected, supplemented with sample buffer, and incubated at 95 °C for 5 min. Protein quantification was done with Pierce™ BCA Protein Assay Kits (Thermo Fisher, 23227) according to the manufacturer's instructions. Equal protein amounts of clarified lysates (20 μg) were subsequently separated by sodium dodecyl sulfate (SDS) polyacrylamide gel electrophoresis (165 V, 400 mA, 60 min) and transferred onto polyvinylidene fluoride membranes by electroblotting (100 V, 400 mA, 90 min). Next, the membranes were incubated for 1 h with Li-Cor blocking buffer (Li-Cor, 927-60001) at RT in Tris-buffered saline (TBS), supplemented with 0.1% Tween (BIO-RAD, 1706531). Next, primary antibodies were added (see below and Table 1a) in Li-Cor blocking buffer supplemented with 0.1% Tween, and the membranes were incubated overnight at 4 °C on a shaker. Following washing with TBS-0.1% Tween, the membranes were incubated for 1 h at RT with corresponding IRDye-conjugated secondary antibodies (See below and Table 1b) in Li-Cor blocking buffer supplemented with 1% SDS and 0.1% Tween. The membrane was visualized with the Odessey imaging system.

Uncropped blots can be find in Supplementary Figs. 7–9. Overview of the primary and secondary antibodies used in western blot experiments can be found in Table 1.

### RT-qPCR analysis of gene expression in fibroblasts, cardiomyocytes, and myocardial tissue

RNA was extracted using Nucleospin RNA XS (Macherey-Nagel, 740902.50) according to the manufacturer's instructions. For cDNA synthesis, total RNA was added to a mixture containing buffer with random hexamers and reverse transcriptase enzyme (TaqMan reverse transcription reagents, Applied biosystems, N8080234) after which samples were incubated for 10 min at 25 °C, 30 min at 48 °C, and 5 min at 95 °C. RT-qPCR was performed on QuantStudio 3 Real-time PCR system (Applied Biosystems) using Taqman Universal PCR Master Mix (Applied Biosystems, 4304437) and Taqman primers according to the manufacturer's instructions. Settings were as follows: 2 min at 95 °C followed by 10 min at 95 °C, 40 cycles (45 cycles for ERBB4) of denaturation at 95 °C for 15 s, and 1 min at 60 °C. All reactions were run in duplicate, and all data were normalized against housekeeping genes glyceraldehyde-3-phosphate dehydrogenase (*GAPDH*), beta-actin (*ACTB*), or phosphoglycerate kinase 1 (*PGK1*). Expression levels were calculated using the comparative cycle method and expressed as fold change (FC) to appropriate controls. The following TaqMan primers were used (Thermo Fisher Scientific): GAPDH (Hs02758991_g1, Mm99999915_g1 and Rn01775763_g1), PGK1 (Hs99999906_m1), ID2 (Hs04187239_m1), BMPR1B (Hs01010965_m1), CACNA2D4 (Hs00297782_m1), DUSP2 (Hs00358879_m1), NR4A1 (Hs00374226_m1), SGK1 (Hs00178612_m1), COL1A1 (Mm00801666_g1), COL3A1 (Hs00943809_m1 and Mm00802305_g1), ERBB4 (Hs00955525_m1 and Rn00572447_m1), ACTB (Mm02619580_g1) and ANP (Mm01255747_g1).

### Histology and immunostaining

Hearts were fixed in 4% PFA, paraffin-embedded, and cut into 5-μm sections. Images were acquired with an Olympus BX43 microscope (Olympus Stream Motion Software) or a CELENA® S Digital Imaging System and analyzed with ImageJ 2.14.0 software. Collagen distribution was visualized by Masson's trichrome staining. Cardiac total fibrosis was expressed as the ratio of positively stained fibrotic area (blue) to the total area and perivascular fibrosis as the ratio of fibrotic area to vascular lumen area. Scar size was determined by measuring the scar length over the total heart area. Hypertrophy and vascularisation were visualized via an isolectin-B4/WGA/DAPI staining. Quantification was performed by a person blinded to the treatment protocol.

### Data analysis and statistics

Statistical analysis was performed using GraphPad Prism version 10. Data are presented as the mean value of biological replicates (unless stated otherwise) ± SD. Data was checked for normality using the Shapiro–Wilk normality test. $P$ value < 0.05 was considered the threshold for statistical significance and all exact $P$ values are provided, when possible, within each figure together with the statistical test performed for each experiment. Comparison between groups was

**Table 1 | Overview of the (a) primary and (b) secondary antibodies used in western blot experiments**

| a) | Primary antibody | | | | |
|---|---|---|---|---|---|
| **Antibody** | **Species** | **Dilution** | **Manufacturer** | **Clone name** | |
| GAPDH | Mouse | 1:2,500 | CST-97166 | Primary monoclonal D4C6R | |
| pAKT | Mouse | 1:1,000 | CST-12694 | Primary monoclonal E4U3U | |
| AKT | Rabbit | 1:1,000 | CST-9272 | Primary polyclonal | |
| pERK1/2 | Rabbit | 1:1,000 | CST-9101 | Primary polyclonal | |
| ERK1/2 | Rabbit | 1:1,000 | CST-9102 | Primary polyclonal | |
| ERBB4 | Rabbit | 1:250 | CST-4795 | Primary monoclonal 111B2 | |
| **b)** | **Secondary antibody** | | | | |
| **Channel** | **Species** | **Dilution** | **Manufacturer** | **–** | |
| 680RD | Goat anti-Mouse | 1:10,000 | Li-cor-926-68070 | – | |
| 800RD | Goat anti-Rabbit | 1:10,000 | Li-cor-926-32211 | – | |

Source data are provided as a Source Data file.

*AKT* protein kinase B, *CST* Cell Signaling Technologies, *ERBB4* erythroblastic leukemia viral oncogene homolog 4, *ERK* extracellular signal-regulated kinase, *GAPDH* glyceraldehyde-3-phosphate dehydrogenase, *RD* infrared dye.

performed using unpaired t-test (two-tailed) or one-way ANOVA with Dunnett or Tukey corrections for multiple comparisons. For the survival curve, a Kaplan–Meier test was performed, follow by the log-rank (Mantel–Cox) test.

### Reporting summary

Further information on experimental design is available in the Nature Research Reporting Summary linked to this article.

### Reporting summary

Further information on research design is available in the Nature Portfolio Reporting Summary linked to this article.

### Data availability

All data supporting the findings of this study are available within the paper and its Supplementary Information. Source data are provided with this paper. The RNA sequencing data generated has been deposited in NCBI's Gene Expression Omnibus and is accessible through GEO Series accession numbers GSE256024 for iAM data [https://www.ncbi.nlm.nih.gov/geo/query/acc.cgi?acc=GSE256024] and GSE261219 for HCF data [https://www.ncbi.nlm.nih.gov/geo/query/acc.cgi?acc=GSE261219]. All mass spectrometry data are available as Supplementary File. Source data are provided with this paper.

### Code availability

The code for the determination of the cross-sectional area of cardiomyocytes from fluorescence images and its analysis associated with the current submission is available at https://doi.org/10.5281/zenodo.13309997. Any updates will also be published on Zenodo, and the final DOI cited in the manuscript.

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

## Acknowledgements

We thank Juan Zhang and Minka Bax (Laboratory of Experimental Cardiology, Leiden University Medical Center, Leiden, the Netherlands) for arranging the transfer of iAMs and providing protocols. We also thank Tine Bruyns, Mandy Vermont and Sophie Lyssens (University of Antwerp, Antwerp, Belgium) for technical support. Some icons of BioRender were used in the preparation of Figs. 5a, 6a and 7a (https://BioRender.com). This work was supported by a Geconcerteerde onderzoeksactie grant (GOA, PID36444) of the University of Antwerp; by a Senior Clinical Investigator fellowship (to V.F.S.), a PhD fellowship (to J.MT.C. and C.C.), and research grants of the Fund for Scientific Research Flanders (Application numbers 1842219N, G021019N, G021420N, 1S49323N, and 11PBU24N); VLIR/iBOF Grant 20-VLIR-iBOF-027 (to N.V., V.F.S., H.L.R., and G.W.D.K.).

## Author contributions

V.F.M.S. and G.W.D.K. conceived and designed research; J.MT.C., B.K.G., E.F., S.V.d.B., C.C., J.V.f., M.R.L.T., L.N., performed experiments; J.MT.C., B.K.G., E.F., S.V.d.B., Y.F., J.O. and E.M.W. analyzed data; J.MT.C., B.K.G., E.F., H.L.R., G.W.D.K. and V.F.M.S. interpreted results of experiments;

J.MT.C. and E.F. prepared figures; J.MT.C. and E.F. prepared manuscript; J.MT.C., B.K.G., E.F., S.V.d.B., C.C., J.V.f, E.M.W., B.V.B., A.A.F.D.V., N.V., D.A.P., D.A., H.L.R., H.D.W., G.W.D.K. and V.F.M.S. edited and revised manuscript; J.MT.C., B.K.G., E.F., S.V.d.B., Y.F., C.C., J.V.f., M.R.L.T., J.O., L.N., E.M.W., B.V.B., A.A.F.D.V., N.V., D.A.P., D.A., H.L.R., H.D.W., G.W.D.K. and V.F.M.S. approved final version of manuscript. N.V., D.A.P., H.L.R., G.W.D.K and V.F.M.S. supervised the research. J.MT.C, B.K.G. and E.F. contributed equally to this manuscript.

## Competing interests

Patent "MODULATORS OF ERBB4 IN THE TREATMENT OF DISEASES"; EP20210160742; Inventors: V.F.M.S., G.W.D.K., E.F., H.D.W.
