## [Peer Review file · Nature Communications]

Small molecule-induced ERBB4 activation to treat heart failure

Corresponding Author: Professor Vincent Segers

Version 0:

Reviewer comments:

Reviewer #1

(Remarks to the Author)

NRG1 has achieved marginal success in early clinical trials, but its short plasma half-life, requirement for continuous intravenous infusion, and its binding to multiple ERBBs have limited its clinical application. Based on NRG1-induced ERBB4 dimerization, the authors identified several compounds with a common pharmacophore through a combination of HTS and chemoinformatics. Subsequently, they focused on the most potent compound EF-1 in terms of ERBB4 dimerization potency and efficacy. Although EF-1 has less potency in promoting ERBB4 dimerization compared to NRG1, it has better half-life and specificity than NRG1. Finally, the authors investigated the biological functions of EF-1 in vitro and in vivo, including its effects on cardiac fibrosis, cardiomyocyte cell death, and cardiac hypertrophy. However, this article did not delve into the molecular mechanisms how EF-1 induces ERBB4 dimerization and its downstream signaling pathways. Overall, while this work gains new insights and approaches for discovering a panel of candidate compounds on therapeutic treatment for chronic heart failure, it falls short in elucidating the molecular action of EF-1 and other candidate compounds. Below are my specific concerns and comments:

- 1) The statistical results presented in Figure 2a-b showed that the ratio of p-Akt/Akt and p-ERK/ERK increased after treatment with 32 μ M EF-1. However, the representative Western blots displayed in the figures exhibit marginal changes.
- 2) The data in this work suggest that EF-1 decreased fibroblast collagen production, cardiomyocyte death and hypertrophy through ERBB4. However, the authors have no further data on elucidating the downstream signaling pathways by EF-1. For instance, the sequencing data indicates that EF-1 can downregulate the TGF β and MAPK/ERK signaling pathways in human cardiac fibroblasts. Further experiments are needed to show whether inhibiting these downstream signaling pathways can suppress the effects of EF-1.
- 3) The authors should perform qRT-PCR to verify that EF-1 and NRG1, indeed, downregulated gene expression levels of the TGF β and MAPK/ERK pathways in HCFs in Fig. 3e.
- 4) The authors showed that EF-1 promoted the phosphorylation of AKT and ERK in cardiomyocytes, and the activation of the PI3K-AKT and MAPK/ERK signaling pathways is associated with cardiomyocyte proliferation. Therefore, further experiments are needed to show whether EF-1 can also promote cardiomyocyte proliferation.
- 5) The data in this work propose that EF-1 did not compete with NRG1 for their binding to ERBB4/ERBB4 dimerization and enhances the effect of NRG1 on ERBB4 dimerization. These data suggested that EF-1 and NRG1 might have different binding sites to ERBB4, and thus the authors may consider presenting experimental data on this hypothesis.
- 6) The authors should provide more evidence that EF-1 dose-dependently decreased H₂O₂-induced cardiomyocyte cell death in Fig.4d-f, such as LDH detection, CCK-8 assay and TUNEL staining.

Reviewer #2

(Remarks to the Author)

Heart diseases are among the leading cause of death worldwide. Thus, any improvement in treating heart diseases are of

great relevance. Previously, it has been shown by several independent laboratories that activation of the NRG1/ERBB4 pathway is a promising strategy to treat different heart diseases. Unfortunately, current approaches to translate this strategy, effective in animal models, remains to be sufficiently reproduced in clinical trials. Here, the authors have "hypothesized that small-molecule-induced activation of ERBB4 is feasible and can protect against myocardial cell death and fibrosis". This is important as recombinant NRG1 has a short half-life, requires intravenous administration, and is a non-selective agonist of the ERBB system, thus might activate ERBB3 receptors which could induce or accelerate cancer growth.

General comment:

This is in general a very well controlled study. The authors performed a high-throughput screen and identified a compound that efficiently induced ERBB4 homodimerization. They selected one compound for further analysis, EF-1, and characterized in detail binding pocket, induced signaling pathways and downstream targets. While there were significant overlaps, there were also significant differences. Yet, functional assays showed that similar to NRG1 EF-1 decreased in an ERBB4-dependent manner cell death and hypertrophy in cultured atrial cardiomyocytes and collagen production in cultured human cardiac fibroblasts. Subsequently, the authors showed that EF-1 treatment before onset of disease can prevent, or at least reduce, fibrosis and cell death. Finally, the authors determined whether EF-1 treatment two weeks after MI, thus in contrast to the other experiments clinically relevant, has an effect on disease outcome. Unfortunately, this part of the manuscript is less developed and the differences are borderline significant or even non-significant (see for details below). Importantly, there is no comparison to NRG1 treatment.

Specific major points:

- 1) The most clinically relevant data, aim of the study, are the data in figure 6 regarding the effect of EF-1 treatment 2 weeks post-MI. The authors state in the abstract "Finally, EF-1 improved cardiac function in a mouse model of myocardial infarction (MI)". Yet, the authors provide no data to justify this statement. First of all, there is no statistical significance. Second, there are no longitudinal data to state that EF-1 treatment improved heart function or maintained heart function.
- 2) The analysis of the experiment in figure 6 is rather poor. Usually, data for ejection fraction and similar are shown longitudinal for each mouse before MI, 1 day post MI, day of treatment (here 2 weeks post-MI) and endpoint (here 4 weeks post-MI). It would further be important to assess whether the treatment had an effect on scar size, apoptosis, vascularization, hypertrophy, and so on (especially, as it is at least for NRG1 known that it affects several cell types in the heart).
- 3) The authors have analyzed echography data before MI and 1, 2, 3, and 4 weeks post MI but failed to describe and discuss these data. As these data are the most important data they need to be included in figure 6. A quick look at the data suggests that EF-1 treatment maintains function (2w: 27.4% and 4w: 33.9% compared to MI: 2w: 29.0% and 4w 23.6%) but does not improve function as stated (and changes in EF are not statistically significant).
- 4) As the authors want to find a compound that has similar effects as NRG1, it is unclear why NRG1 was not included in the in vivo experiments as control.

Specific minor points:

- 1) In Figure 2b, it appears 15 min treatment with Veh results in decreased phosphorylation of ERK1/2, while EF-1 stimulation maintains the levels of ERK1/2 phosphorylation. Yet, the authors stated "EF-1 also induced a transient phosphorylation of ERK1/2 (Fig. 2b)". Please clarify.
- 2) Signaling experiments should be performed in parallel with NRG1 to evaluate how efficient EF-1 induces signaling compared to NRG1.
- 3) ERBB4 knockdowns were with ~50 and 60% rather inefficient, however, the authors still observed a strong inhibition of the effects of EF-1. Please explain. Furthermore, the authors state "Next, EF-1 dose-dependently decreased H₂O₂-induced cardiomyocyte cell death with more than 75% at its highest dose, an effect that was blunted by ErbB4 knockdown (Fig. 4f)." Yet, H₂O₂-induced cardiomyocyte cell death was not completely rescued and thus "blunted" might be misleading. Note, CPD is not explained and not listed under abbreviations (it is only once explained in Figure S3).
- 3) It appears that iAMs are of very similar size and thus induction of hypertrophy can be easily assessed. This is not the case for neonatal rat cardiomyocytes. Is it normal that iAMs grow as individual isolated cells not touching neighboring cells? This might be helpful, as one usually has issues with determining hypertrophy in 2D as one needs to consider also the height of the cells. Please comment; maybe cite studies that previously evaluated hypertrophy in iAMs.
- 4) Authors need to provide all data ("data not shown").
- 5) In Figure 5a, the "B", "C", "D", and "E-G" should most likely not be capitalized. Same applies to Figure 6.
- 6) In Figure 6e, total fibrosis is barely visible. Please provide close ups that clearly show an increase in interstitial fibrosis upon AngII treatment.
- 8) Please note, that the role of NRG1/ERBB4 in heart regeneration PMID: 19632177 has been challenged PMID: 25848746.
- 9) Authors need to provide the approval number of their animal study.

Reviewer #3

(Remarks to the Author)

This manuscript describes identification of small molecule activators of ErbB4 with a high throughput screen of a large library of compounds, and using a variety of in vitro and in vivo models demonstrates that the selected compound is able to recapitulate cardioprotective and antifibrotic effects of recombinant neuregulin-1. The potential for the NRG/ErbB signaling system to be a target of therapy in heart disease is not novel. However this represents a significant advance to this field as it represents the first demonstration of a small molecule activator of this pathway, which has therapeutic potential.

It appears that the small molecule selected is also able to induce ErbB3/ErbB2 heterodimer formation based upon a ErbB3/ErbB2 screen. Whether it also induces ErbB4/ErbB3 heterodimer formation is not specifically assessed, but seems

likely given that it activates both ErbB3 and ErbB4 heterodimers with ErbB2. There is also no assessment of its effects on ErbB1 dimerization. Additional experimental work would be helpful to more completely understand the actions of the selected compounds.

While the ErbB4 silencing experiments and work with the ErbB4 knockout all support an activity in the selected assays that involves ErbB4, ErbB3 is also present in the cells, as well as the in vivo experimental systems selected. As the small molecule activator of ErbB4 also activates ErbB3, experimental studies addressing the effect of ErbB3 silencing would be needed to fully address the potential role of ErbB3 activation in these experimental systems, and fully support the conclusions as written throughout the manuscript.

It would be interesting to know if the library contained compounds active in the ErbB3/ErbB2 assay, but did not activate ErbB4. Study of such a compound in the select assays might serve as an interesting comparison.

It was not clear whether the effect of gender was examined in the in vivo work. This would seem important as there have been some instances where the neuregulin/ErbB signaling system has distinct effects in male vs. female mice (e.g. H. Yin et al, JMCC 2021, PMID: 33259856).

Version 1:

Reviewer comments:

Reviewer #1

(Remarks to the Author)

The authors have addressed my major comments, so I suggest it is suitable for publication in NC.

Reviewer #2

(Remarks to the Author)

The authors have taken all my comments seriously and addressed all of them satisfactorily by adding a large amount of new data.

Reviewer #3

(Remarks to the Author)

Thank you for the additional experimental work. The results add considerable value to the manuscript.

Response Letter

Manuscript Title: Small molecule-induced ERBB4 activation to treat heart failure

Dear Reviewers,

We would like to thank the reviewers for their comments and suggestions on our manuscript. Based on these comments, we have performed additional experiments and have revised the manuscript accordingly. Here, we provide a response to each comment.

Reviewer #1 comments:

NRG1 has achieved marginal success in early clinical trials, but its short plasma half-life, requirement for continuous intravenous infusion, and its binding to multiple ERBBs have limited its clinical application. Based on NRG1-induced ERBB4 dimerization, the authors identified several compounds with a common pharmacophore through a combination of HTS and chemoinformatics. Subsequently, they focused on the most potent compound EF-1 in terms of ERBB4 dimerization potency and efficacy. Although EF-1 has less potency in promoting ERBB4 dimerization compared to NRG1, it has better half-life and specificity than NRG1. Finally, the authors investigated the biological functions of EF-1 in vitro and in vivo, including its effects on cardiac fibrosis, cardiomyocyte cell death, and cardiac hypertrophy. However, this article did not delve into the molecular mechanisms how EF-1 induces ERBB4 dimerization and its downstream signaling pathways. Overall, while this work gains new insights and approaches for discovering a panel of candidate compounds on therapeutic treatment for chronic heart failure, it falls short in elucidating the molecular action of EF-1 and other candidate compounds. Below are my specific concerns and comments:

We thank the reviewer for their careful evaluation of our manuscript.

Comment 1

The statistical results presented in Figure 2a-b showed that the ratio of p-Akt/Akt and p-ERK/ERK increased after treatment with 32 μ M EF-1. However, the representative Western blots displayed in the figures exhibit marginal changes.

We agree with the reviewer that the changes in the western blots are subtle. Nevertheless, the statistical analysis was performed on data obtained from four independent experiments, each consisting of separate western blots. The results were highly reproducible, and the observed changes, although subtle, were consistent across all experiments. To further clarify this, we have included a note in the figure legend.

Changes made to the manuscript:

Figure legend referring to Figure 2b.

"Figure 2. EF-1 and NRG1 activate similar downstream signaling pathways in cultured cardiomyocytes. (a-b) Representative western blot images showing the effect of **(a)** NRG1 (10 nM) and **(b)** EF-1 (32 μ M) on pAKT/AKT and pERK1/2/ERK1/2 pathways in iAMs stimulated for different times. Bar graphs show the average effects of three or four independent experiments \pm SD; one-way ANOVA with Dunnett's multiple comparisons test. Despite the subtle changes in the representative blots of EF-1, the results were highly reproducible."

Comment 2

The data in this work suggest that EF-1 decreased fibroblast collagen production, cardiomyocyte death and hypertrophy through ERBB4. However, the authors have no further data on elucidating the downstream signaling pathways by EF-1. For instance, the sequencing data indicates that EF-1 can downregulate the TGFbeta and MAPK/ERK signaling pathways in human cardiac fibroblasts. Further experiments are needed to show whether inhibiting these downstream signaling pathways can suppress the effects of EF-1.

The reviewer asks us to test the role of downstream ERBB4 signaling on phenotypic changes induced by EF-1. Therefore, we performed additional studies with inhibitors of the PI3/AKT pathway (LY294002) and of the MEK1/2 pathway (PD98059) and tested whether these inhibitors affected the phenotypic responses induced by EF-1.

Changes made to the manuscript:

Results

Page 7, lines 146-152

"As shown in Figure 2b, EF-1, as NRG1, signals through the AKT- and ERK1/2-pathway.¹ To determine if the observed cardiomyocyte survival and anti-hypertrophic effects of EF-1 are AKT- and ERK1/2-pathway related, we repeated the experiments with pre-incubation of iAMs with LY294002 (a PI3K/AKT-inhibitor), and PD98059 (a MEK1/2-inhibitor). We observed that pretreatment with LY294002 blocked the pro-survival effect of EF-1 on iAMs treated with H₂O₂ (Fig. 4h), suggesting that that the PI3K/Akt pathway contributes to the cell survival capacities of EF-1. In addition, we observed that the anti-hypertrophic effect of EF-1 was blocked by inhibition of the ERK1/2 pathway (Fig. 4i)."

Methods

Page 22, lines 497-503

"In vitro pathway inhibitors"

LY294002 (MedChem Express, HY-10108) was used to inhibit the PI3K/AKT pathway. iAMs were treated with EF-1 (32 μ M) or vehicle, with or without 10 minutes pretreatment with LY294002 (20 μ M) and cell death was induced with H₂O₂.

PD98059 (MedChem Express, HY-12028) was used to inhibit the ERK1/2 pathway. iAMs were treated with EF-1 (32 μ M) or vehicle, with or without 120 minutes pretreatment with PD98059 (50 μ M) and cardiomyocyte hypertrophy was induced with AngII."

Figure 4

“Figure 4. EF-1 decreases collagen production in fibroblasts, and decreases cardiomyocyte cell death and hypertrophy. (g) CSA of iAMs after AngII exposure in the presence or absence of EF-1 (4–32 μM). **(h)** Effect of H₂O₂ on total cell death of iAMs in the presence or absence of the PI3K/AKT-pathway inhibitor LY294002, with and without EF-1 (32 μM). **(i)** CSA of iAMs after AngII exposure in the presence or absence of the ERK-pathway inhibitor PD98059, with and without EF-1 (32 μM). All data are represented as mean ± SD, n = 3–14 in each group, unpaired t-test or one-way ANOVA with Dunnett’s multiple comparisons test against *siScr*. P-values shown are between EF-1 + *siScr* and EF-1 + *siERRB4*. Scale bar = 100 μm, n = 20 CSA in each group, one-way ANOVA with Tukey’s multiple comparisons test. All data are represented as mean ± SD. AngII, angiotensin II; ANOVA, analysis of variance; COL3A1, collagen type 3 alpha 1; CSA, cross sectional area; FC, fold change; H₂O₂, hydrogen peroxide; HCF, human

c

a

r

d

Comment 3

The authors should perform qRT-PCR to verify that EF-1 and NRG1, indeed, downregulated gene expression levels of the TGFβ and MAPK/ERK pathways in HCFs in Fig. 3e.

As suggested by the reviewer, we performed qRT-PCR to verify the downregulation of gene expression levels of the TGFβ, MAPK/ERK and PI3k-Akt pathways by EF-1 and NRG1 in HCFs. For each pathway, we selected the two most downregulated genes shared by both NRG1 and EF-1 in our sequencing data.

r

o

b

l

a

s

Changes made to the manuscript:

Results

Page 6, lines 115-118

“To confirm the RNA sequencing data, we performed qRT-PCR analysis on two genes from each signaling pathway, i.e. those with the largest decrease in expression induced by both EF-1 and NRG1. All six genes, belonging to either the TGF- β (Fig. 3f), MAPK/ERK (Fig. 3g) or PI3k-AKT pathway (Fig. 3h) showed a significant decrease in expression when HCFs were treated with EF-1 or NRG1.”

Methods

Page 28, lines 641-646

“The following TaqMan primers were used (Thermo Fisher Scientific): GAPDH (Hs02758991_g1, Mm99999915_g1 and Rn01775763_g1), PGK1 (Hs99999906_m1), ID2 (Hs04187239_m1), BMPR1B (Hs01010965_m1), CACNA2D4 (Hs00297782_m1), DUSP2 (Hs00358879_m1), NR4A1 (Hs00374226_m1), SGK1 (Hs00178612_m1), COL1A1 (Mm00801666_g1), COL3A1 (Hs00943809_m1 and Mm00802305_g1), ERBB4 (Hs00955525_m1 and Rn00572447_m1), ACTB (Mm02619580_g1) and ANP (Mm01255747_g1).”

Figure 3

“Figure 3. EF-1 and NRG1 activate similar downstream signaling pathways in human cultured cardiac fibroblasts. (f-h) RT-qPCR on signaling pathway genes in HCFs; normalized FC compared to vehicle: (f) ID2 and BMPR1B, genes involved in TGF- β signaling, (g) CACNA2D4 and DUSP2, genes involved in MAPK/ERK signaling and (h) NR4A1 and SGK1, genes involved in PI3k-Akt signaling. All data are represented as mean \pm SD, n = 4 in each group, one-way ANOVA with Tukey’s multiple comparisons test. AKT, protein kinase B; BMPR1B, bone morphogenetic protein receptor type 1B; CACNA2D4, calcium voltage-gated channel auxiliary subunit alpha-2 delta-4; DEG, differentially expressed gene; DUSP2, dual specificity phosphatase 2; ERK, extracellular signal-regulated kinase; FC, fold change; GSEA, gene set enrichment analysis; HCF, human cardiac fibroblast; ID2, Inhibitor of DNA Binding 2; MAPK, mitogen-activated protein kinase; NES, normalized enrichment score; NRG1, neuregulin-1; NR4A1, nuclear receptor subfamily 4 group A member 1; PI3K, phosphatidylinositol 3-kinase; RT-qPCR, real-time quantitative polymerase chain reaction; SGK1, serum/glucocorticoid-regulated kinase 1; TGF- β , transforming growth factor- β .”

Comment 4

The authors showed that EF-1 promoted the phosphorylation of AKT and ERK in cardiomyocytes, and the activation of the PI3K-AKT and MAPK/ERK signaling pathways is associated with cardiomyocyte

proliferation. Therefore, further experiments are needed to show whether EF-1 can also promote cardiomyocyte proliferation.

We value the reviewer's suggestion to investigate whether EF-1 promotes cardiomyocyte proliferation. To assess cell proliferation, we performed EdU (5-ethynyl-2'-deoxyuridine) staining on differentiated and proliferating immortalized atrial myocytes (iAMs).

Changes made to the manuscript:

Results

Page 7, lines 153-157

“EF-1, as NRG1, did not promote iAM proliferation

Because NRG1 promotes cardiomyocyte proliferation,^{2,3} although contested,⁴ we tested if EF-1 could promote iAM proliferation. We performed an 5-ethynyl-2'-deoxyuridine (EdU)-staining on proliferating iAMs, stimulated with either EF-1 or NRG1, and observed a significant decrease in EdU-positive nuclei in both groups compared to the vehicle group (Suppl. Figure 3c).”

Methods

Page 23, lines 504-513

“EdU proliferation assay

EdU proliferation assay was performed using the Click-iT™ EdU Cell Proliferation Kit for Imaging (ThermoFisher Scientific, C10337) following the protocol. Proliferating iAMs were seeded at a density of 10⁵ cells/well in proliferating medium in 24-well plates and incubated overnight. The cells were stimulated with either vehicle, EF-1 (32 μM) or NRG1 (0.1 μM) for 24 h at 37°C. All wells contained DMSO at a final concentration of 0.9%. Afterwards, the cell culture media were replaced with 10 μM EdU in proliferating medium and incubated for 24h at 37°C. Next, iAMs were fixed with 4% PFA for 30 min at 4°C. Cells were washed thrice with PBS, permeabilized by incubation with 0.1% Triton X-100 for 10 min at RT and counterstained with DAPI. Images were obtained by fluorescence microscopy (Celena S).”

Supplementary Figure 3

“Supplementary Figure 3. Receptor binding assay and proliferation properties of EF-1. (c) EdU-positive cells (FC to Veh) in iAMs, stimulated with EF-1 or NRG1. All data are presented as mean ± SD, n = 5 in each group, one-way ANOVA test with Dunnett’s multiple comparison test. CPD, compound;

DRC; dose response curve; EdU, 5-Ethynyl-2'-deoxyuridine; FC, fold change; F-NRG1, fluorescently labeled NRG1; M, molar concentration; NRG1, neuregulin-1; Veh, vehicle.”

Comment 5

The data in this work propose that EF-1 did not compete with NRG1 for their binding to ERBB4/ERBB4 dimerization and enhances the effect of NRG1 on ERBB4 dimerization. These data suggested that EF-1 and NRG1 might have different binding sites to ERBB4, and thus the authors may consider presenting experimental data on this hypothesis.

We agree that it would be ideal to know the binding site of EF-1 on ERBB4. Our current data demonstrate that EF-1 does not compete with NRG1 for binding to ERBB4, suggesting that they do not share the same binding site. At this point, we have no experimental showing the exact binding site of EF-1 on the ERBB4 receptor. To identify the binding site, elaborate studies with crystallography or cryo-electron microscopy would be necessary; two methods with high technical risks and requiring extensive resources.

Comment 6

The authors should provide more evidence that EF-1 dose-dependently decreased H₂O₂-induced cardiomyocyte cell death in Fig.4d-f, such as LDH detection, CCK-8 assay and TUNEL staining.

We performed a WST-1 assay (similar to the CCK-8 assay) and TUNEL staining on immortalized atrial myocytes (iAMs) stimulated with a single dose EF-1 and H₂O₂, as suggested by the reviewer.

Changes Made in the Manuscript:

Results

Pages 6-7, lines 137-143

“The protective effects of EF-1 on cardiomyocyte viability were further confirmed by WST-1 and TUNEL assay. Treatment with 100 μM H₂O₂ reduced iAM cell viability, as indicated by decreased WST-1 absorbance, but pre-treatment with EF-1 significantly increased cell viability (Suppl. Fig. 4a). H₂O₂ also significantly increased the number of TUNEL-positive (apoptotic) cells, and EF-1 also significantly reduced this number (Suppl. Fig. 4b).”

Methods

Page 21, lines 454-467

“To confirm that EF-1 decreased H₂O₂-induced cell death, we repeated the experiment with two different assays (WST-1 and TUNEL assay).

Following the transfer of 150 μL of culture medium for the ToxiLight assay, 7 μL of WST-1 reagent (Merck, 5015944001) was added directly to the remaining cell culture medium in each well. The cells were incubated at 37°C for an additional 4 hours. After incubation, the absorbance of the samples was measured at 450 nm (690 nm was used as reference wavelength and subtracted) using a microplate reader (Epoch).

To assess apoptosis, a TUNEL assay was performed using the In Situ Cell Death Detection Kit, Fluorescein (Merck, 11684795910) according to the manufacturer's instructions. After 4-hour incubation, the cells were fixed with 4% paraformaldehyde for 15 minutes at room temperature, followed by permeabilization with 0.1% Triton X-100 in 0.1% sodium citrate for 2 minutes on ice. Cells were then incubated with the TUNEL reaction mixture in a humidified chamber at 37°C for 1 hour in the dark. After washing with PBS, the samples were analysed using fluorescence microscopy to detect apoptotic cells."

Supplementary Figure 4

“Supplementary Figure 4. EF-1 decreases H₂O₂-induced cardiomyocyte cell death. (a) Effect of EF-1 on cell viability assessed with the WST-1 assay. **(b)** Effect of EF-1 on TUNEL staining (red). The corresponding bar graph shows the average percentage of TUNEL-positive cells. All data are represented as mean ± SD, n = 9-20 in each group, one-way ANOVA with Tukey's multiple comparisons test. ANOVA, analysis of variance; FC, fold change; H₂O₂, hydrogen peroxide; iAM, conditionally immortalized rat atrial myocyte; TUNEL, terminal deoxynucleotidyl transferase dUTP nick end labelling; Veh, vehicle; WST-1, 2-(4-iodophenyl)-3-(4-nitrophenyl)-5-(2,4-disulfophenyl)-2H-tetrazolium.”

Reviewer #2 comments:

Heart diseases are among the leading cause of death worldwide. Thus, any improvement in treating heart diseases are of great relevance. Previously, it has been shown by several independent laboratories that activation of the NRG1/ERBB4 pathway is a promising strategy to treat different heart diseases. Unfortunately, current approaches to translate this strategy, effective in animal models, remains to be sufficiently reproduced in clinical trials. Here, the authors have “hypothesized that small-molecule-induced activation of ERBB4 is feasible and can protect against myocardial cell death and fibrosis”. This is important as recombinant NRG1 has a short half-life, requires intravenous administration, and is a non-selective agonist of the ERBB system, thus might activate ERBB3 receptors which could induce or accelerate cancer growth.

General comment:

This is in general a very well controlled study. The authors performed a high-throughput screen and identified a compound that efficiently induced ERBB4 homodimerization. They selected one compound for further analysis, EF-1, and characterized in detail binding pocket, induced signaling pathways and downstream targets. While there were significant overlaps, there were also significant differences. Yet, functional assays showed that similar to NRG1 EF-1 decreased in an ERBB4-dependent manner cell death and hypertrophy in cultured atrial cardiomyocytes and collagen production in cultured human cardiac fibroblasts. Subsequently, the authors showed that EF-1 treatment before onset of disease can prevent, or at least reduce, fibrosis and cell death. Finally, the authors determined whether EF-1 treatment two weeks after MI, thus in contrast to the other experiments clinically relevant, has an effect on disease outcome. Unfortunately, this part of the manuscript is less developed and the differences are borderline significant or even non-significant (see for details below). Importantly, there is no comparison to NRG1 treatment.

We thank the reviewer for their careful evaluation of our manuscript.

Comment 1

The most clinically relevant data, aim of the study, are the data in figure 6 regarding the effect of EF-1 treatment 2 weeks post-MI. The authors state in the abstract “Finally, EF-1 improved cardiac function in a mouse model of myocardial infarction (MI)”. Yet, the authors provide no data to justify this statement. First of all, there is no statistical significance. Second, there are no longitudinal data to state that EF-1 treatment improved heart function or maintained heart function.

We thank the reviewer for their detailed feedback regarding the clinical relevance of our data. In response to the concern that the data do not show statistical significance and there are no longitudinal data to support the claim of improved cardiac function, we have updated the graph on ejection fraction, left-ventricular end-diastolic volume and left-ventricular end-systolic volume to include longitudinal data for each male and female mouse, showing measurements taken before MI, 2 weeks post-MI, 3 weeks post-MI, and at the endpoint (5 weeks post-MI).

Changes made to the manuscript:

Results

Page 9, lines 204-209

“Representative echocardiographic images of all groups in female mice are shown in Fig. 7b. In Veh-treated female mice, MI significantly increased left ventricular end-diastolic volume (LVEDV; Fig. 7c) and left ventricular end-systolic volume (LVESV; Fig. 7d) and decreased left ventricular ejection fraction

(EF, Fig. 7e). Four-week treatment with EF-1 significantly attenuated the increase in LVEDV and LVESV and resulted in a trend towards an increased EF ($p = 0.0858$). In male mice, EF-1 did not affect LVEDV, LVESV, or EF (Fig. 7c, 7d, 7e).”

Figure 7

“Figure 7. EF-1 decreases ventricular dilation after MI in female mice. (b) Representative echocardiographic images in end-diastole and end-systole of sham-operated control female mice and of mice that underwent MI for 4 weeks with EF-1 or Veh. Graphs showing (c) LVEDV, (d) LVESV and (e) EF in the different experimental groups during the entire experiment in both female and male mice.”

Comment 2

The analysis of the experiment in figure 6 is rather poor. Usually, data for ejection fraction and similar are shown longitudinal for each mouse before MI, 1 day post MI, day of treatment (here 2 weeks post-MI) and endpoint (here 4 weeks post-MI). It would further be important to assess whether the treatment had an effect on scar size, apoptosis, vascularization, hypertrophy, and so on (especially, as it is at least for NRG1 known that it affects several cell types in the heart).

As suggested by the reviewer, we added longitudinal data for each mouse (see comment 1) and performed additional analyses for scar size, apoptosis, vascularisation and hypertrophy and included the data in the revised manuscript as shown below.

Changes Made in the Manuscript:

Results

Pages 9-10, lines 199-220

“In view of the above effects induced by EF-1, and also because NRG1 has been shown to attenuate adverse cardiac remodeling in MI models, we assessed the effects of EF-1 in a murine MI model (Fig.

7a).⁵⁻⁷ Male and female mice were randomized 1 week after ligating the left anterior descending artery (LAD) to implantation of osmotic minipumps with either EF-1 or Veh. Treatment lasted for 28 days, during which cardiac size and function was evaluated using echocardiography (Suppl. Table 1e, 1f). Representative echocardiographic images of all groups in female mice are shown in Fig. 7b. In Veh-treated female mice, MI significantly increased left ventricular end-diastolic volume (LVEDV; Fig. 7c) and left ventricular end-systolic volume (LVESV; Fig. 7d) and decreased left ventricular ejection fraction (EF, Fig. 7e). Four-week treatment with EF-1 significantly attenuated the increase in LVEDV and LVESV and resulted in a trend towards an increased EF ($p = 0.0858$). In male mice, EF-1 did not affect LVEDV, LVESV, or EF (Fig. 7c, 7d, 7e).

Hypertrophy and vascularization in heart tissue were visualized using an isolectin-B4/Wheat Germ Agglutinin (WGA)/4',6-diamidino-2-phenylindole dihydrochloride (DAPI) staining. There was a trend to increase in cardiomyocyte cross-sectional area in the MI/Veh and a significant increase in cross-sectional area in the MI/EF-1 group, in both male and female mice (Fig. 7f). Additionally, a marked decrease in number of capillaries per area (μm^2) of the heart was detected in both MI/Veh and MI/EF-1 groups in female mice, but not in male mice (Fig. 7g). This decrease was not observed when the number of capillaries was normalized to cardiomyocytes, in both sexes (Fig. 7h).

Interstitial fibrosis in the remote myocardium of the infarcted hearts was significantly lower in female mice treated with EF-1, but not in male mice (Fig. 7i; 7j). EF-1 did not affect scar size in female or male mice (Fig. 7k)."

Methods

Page 28-29, lines 648-655

"Images were acquired with an Olympus BX43 microscope (Olympus Stream Motion Software) or a CELENA® S Digital Imaging System and analysed with ImageJ 2.14.0 software. Collagen distribution was visualized by Masson's trichrome staining. Cardiac total fibrosis was expressed as the ratio of positively stained fibrotic area (blue) to the total area and perivascular fibrosis as the ratio of fibrotic area to vascular lumen area. Scar size was determined by measuring the scar length over the total heart area. Hypertrophy and vascularisation were visualized via an WGA/isolectin-B4/DAPI staining. All quantifications were performed by a person blinded to the treatment protocol."

Figure 7

“Figure 7. EF-1 decreases ventricular dilation after MI in female mice. (a) Design of the *in vivo* MI experiment. **(b)** Representative echocardiographic images in end-diastole and end-systole of sham-operated control female mice and of mice that underwent MI for 4 weeks with EF-1 or Veh. Graphs showing **(c)** LVEDV, **(d)** LVESV and **(e)** EF in the different experimental groups during the entire

experiment in both female and male mice. **(f)** CSA of cardiomyocytes analyzed with an isolectin-B4/WGA/DAPI staining in female and male mice. **(g-h)** Vascularization analysis of capillaries with an isolectin-B4/WGA/DAPI staining, showing the number of capillaries in female and male mice **(g)** per area of cardiomyocytes and **(h)** per 100 cardiomyocytes. **(i)** Representative images of Masson's trichrome staining of interstitial fibrosis in the remote zone in female mice. **(j)** Corresponding graph quantifying the fibrotic area in the different experimental groups in female mice and the graph of the quantification of the fibrotic area in male mice. **(n)** Scar size analysis of the myocardial infarction area in female and male mice. Data are represented as mean \pm SD, n = 8–10 in each group, one-way ANOVA test with Tukey's correction for multiple testing. ANOVA, analysis of variance; CSA, cross-sectional area; Ctrl, control; EF, ejection fraction; IP, intraperitoneal; LVEDV, left-ventricular end-diastolic volume; LVEDS, left-ventricular end-systolic volume; MI, myocardial infarction; SD, standard deviation; Veh, vehicle; WGA, wheat germ agglutinin."

Comment 3

The authors have analyzed echography data before MI and 1,2, 3, and 4 weeks post MI but failed to describe and discuss these data. As these data are the most important data they need to be included in figure 6. A quick look at the data suggests that EF-1 treatment maintains function (2w: 27.4% and 4w: 33.9% compared to MI: 2w: 29.0% and 4w 23.6%) but does not improve function as stated (and changes in EF are no statistically significant).

In our revised manuscript, we have included the longitudinal echography data at the specified time points (before MI, and 3, 4, and 5 weeks post-MI) in Figure 7.

Changes Made in the Manuscript: see comment 1.

Comment 4

As the authors want to find a compound that has similar effects as NRG1, it is unclear why NRG1 was not included in the in vivo experiments as control.

NRG1 has been shown to be cardioprotective in multiple studies, mainly by activating the ERBB4 receptor. Nevertheless, our primary aim was not to mimic NRG1, but to demonstrate feasibility of identification of selective ERBB4 small-molecule agonists. Next, we studied the potential therapeutic benefits of EF-1 as a selective ERBB4 agonist. We agree that NRG1 could be included as a positive control, but because of the differences in pharmacokinetics, receptor selectivity and downstream signaling pathways, we opted to focus on experiments using ERBB4 KO mice, instead of including NRG1 as a control.

Changes made to the manuscript: we have clarified this point in discussion section of the manuscript.

Discussion

Page 11, lines 222-223

"In this study, we aimed to investigate the effects of ERBB4 small-molecule agonists on cardiac function and survival."

Comment 5

In Figure 2b, it appears 15 min treatment with Veh results in decreased phosphorylation of ERK1/2, while EF-1 stimulation maintains the levels of ERK1/2 phosphorylation. Yet, the authors stated “EF-1 also induced a transient phosphorylation of ERK1/2 (Fig. 2b)”. Please clarify.

In our experiments, we observed that vehicle (DMSO) treatment alone influenced the phosphorylation levels of ERK1/2, as evidenced by the decreased phosphorylation seen over time. DMSO is known to inhibit phosphorylation of a number of signaling pathways. For this reason, we added a vehicle control at each time point, and not just at time zero (which could lead to the conclusion that a compound dissolved in DMSO decreases ERK phosphorylation).

The statement that "EF-1 also induced a transient phosphorylation of ERK1/2 (Fig. 2b)" refers to the specific response elicited by EF-1 treatment compared to its corresponding vehicle control at each time point, rather than a comparison to baseline or other conditions.

Comment 6

Signaling experiments should be performed in parallel with NRG1 to evaluate how efficient EF-1 induces signaling compared to NRG1.

We thank the reviewer for this suggestion. In response to this comment, we performed western blot analysis for NRG1 under the same experimental conditions as EF-1.

Changes Made in the Manuscript:

Results

Page 5, lines 88-99

“Canonical pathways activated by NRG1 in cardiomyocytes are the phosphatidylinositol 3-kinase (PI3K)/protein kinase B (AKT) and mitogen-activated protein kinase (MAPK)/ extracellular signal-regulated kinase (ERK) pathways.⁸⁻¹⁰ Phosphorylation of AKT and ERK1/2 was assessed by western blot analysis in conditionally immortalized rat atrial myocytes (iAMs).¹¹ Compared to cardiomyocytes derived from induced pluripotent stem cells, iAMs are easier to culture and show a more complete and reliable cardiomyogenic differentiation.¹¹ As shown in Figure 2a, NRG1 showed a significant increase in phosphorylated AKT levels after already 15 minutes. This phosphorylation decreased over time. Additionally, NRG1 showed a time-dependent phosphorylation of ERK1/2 (Fig. 2a). This experiment was repeated with EF-1 on both pathways. Compound EF-1 induced a time-dependent phosphorylation of AKT (Fig. 2b) in cardiomyogenically differentiated iAMs. EF-1 also induced a transient phosphorylation of ERK1/2 (Fig. 2b).

Discussion

Page 13, lines 273-275

“ERK1/2 phosphorylation induced by EF-1 is transient with peak levels at 15 min, which is in line with published data and our data on NRG1-induced ERK1/2 phosphorylation in neonatal rat ventricular myocytes.^{12”}

Methods

"iAMs were seeded at a density of 10^6 cells/well in a transparent 6-well plate (Greiner Bio-One, 657160) and treated with either NRG1 (10nM), EF-1 (32 μ M) or PBS for 0, 15, 60, 120, or 240 min; all wells contained DMSO at a final concentration of 0.9%."

Figure 2

"Figure 2. EF-1 and NRG1 activate similar downstream signaling pathways in cultured cardiomyocytes. (a-b) Representative western blot images showing the effect of (a) NRG1 (10 nM) and (b) EF-1 (32 μ M) on pAKT/ACT and pERK1/2/ERK1/2 pathways in iAMs stimulated for different times. Bar graphs show the average effects of three or four independent experiments \pm SD; one-way ANOVA with Dunnett's multiple comparisons test. Despite the subtle changes in the representative blots of EF-1, the results were highly reproducible."

Comment 7

ERBB4 knockdowns were with ~50 and 60% rather inefficient, however, the authors still observed a strong inhibition of the effects of EF-1. Please explain. Furthermore, the authors state "Next, EF-1 dose-dependently decreased H2O2-induced cardiomyocyte cell death with more than 75% at its highest dose, an effect that was blunted by Erbb4 knockdown (Fig. 4f)." Yet, H2O2-induced cardiomyocyte cell death was not completely rescued and thus "blunted" might be misleading. Note, CPD is not explained and not listed under abbreviations (it is only once explained in Figure S3).

Despite these suboptimal knockdown efficiencies, we observed a strong inhibition of EF-1's effects in our experiments. Even with a partial knockdown efficiency, the remaining ERBB4 expression may still be sufficient to mediate some level of EF-1 signaling. ERBB4 signaling can be complex, and even a moderate reduction in receptor expression may significantly impact downstream signaling pathways. We discussed this matter in the discussion of the revised manuscript.

Changes to the manuscript: we added a hypothesis to the discussion section about the ERBB4 knockdown experiments, changed to word 'blunted' to 'diminished' and added the abbreviation CPD to the abbreviation list.

Results

Page 6, lines 135-136

"Next, EF-1 dose-dependently decreased H₂O₂-induced cardiomyocyte cell death with more than 75% at its highest dose, an effect that was diminished by *ErbB4* knockdown (Fig. 4f)."

Discussion

Page 13, lines 286-287

"Additionally, ERBB4 signaling is complex, and even partial ERBB4 *in vitro* knockdown and *in vivo* knock-out experiments may significantly impact downstream signaling pathways."

Comment 8

It appears that iAMs are of very similar size and thus induction of hypertrophy can be easily assessed. This is not the case for neonatal rat cardiomyocytes. Is it normal that iAMs grow as individual isolated cells not touching neighboring cells? This might be helpful, as one usually has issues with determining hypertrophy in 2D as one needs to consider also the height of the cells. Please comment; maybe cite studies that previously evaluated hypertrophy in iAMs.

We thank the reviewer for this comment.

The growth of cardiomyocytes as individual, isolated cells not touching neighbouring cells is a characteristic observed in many *in vitro* culture systems. This pattern can be advantageous for studying hypertrophy in a 2D setting, as it allows for more accurate measurements of cell size and shape, which are crucial parameters in assessing hypertrophic responses. Furthermore, it is important to measure cell size and shape of each individual cell without an effect of neighbouring cells and their possible interaction or influence.

Previous studies have not yet evaluated hypertrophy in iAMs as they are only recently developed/made but these studies use other cardiomyocytes to assess the same experimental set-up. For instance, research by Watkins S et al. demonstrated hypertrophic responses in cardiomyocytes induced by angiotensin-II and the same data analysis used as us, confirming their utility in studying cardiac hypertrophy mechanisms *in vitro*.^{13,14}

Comment 9

Authors need to provide all data ("data not shown").

As requested by the reviewer, we included more graphs in Supplementary Figure 6 to accompany Figure 5 in the manuscript, along with a description in the results section.

Results

Page 8, lines 174-175

“NA-1 did not significantly prevent interstitial or perivascular AngII-induced fibrosis (Suppl. Fig. 6a).”

Page 8, lines 180-186

“Since the commonly used AngII dose of 1,000 ng/kg/day induced 80% mortality in the *ErbB4*-null mice (Suppl. Fig. 6c), the dose of AngII was lowered to 400 ng/kg/day,¹⁵ EF-1 did not significantly prevent interstitial or perivascular AngII-induced fibrosis in *ErbB4*-null mice indicating that ERBB4 is necessary for the effects of EF-1 on cardiac fibrosis (Fig. 5g). Of note, when *iCAGCre-ErbB4^{fl/fl}* mice were injected with corn oil without tamoxifen, the inhibitory effects of EF-1 on AngII-induced interstitial and perivascular fibrosis were preserved (Suppl. Fig. 6d).”

Supplementary Figure 6

“Supplementary Figure 6. Effect of AngII in male WT mice with compound NA-1 and the validation of genotype and phenotype of iCAGCre-ErbB4 f/f mice. (a) Representative images of Masson’s trichrome staining of AngII-induced myocardial fibrosis following treatment with NA-1 and corresponding graphs showing the quantitation for total and perivascular fibrosis in male mice. **(b)** Representative western blot image and corresponding graph of the validation of the presence of ErbB4 in wild type mice and in iCAGCre-ErbB4^{f/f} mice treated with tamoxifen or corn oil. **(c)** Survival curve depicting the effects of

1000 ng/kg AngII treatment on mouse survival. **(d)** Representative images of Masson's trichrome staining of AngII-induced myocardial fibrosis in *iCAGCre-ErbB4^{ff}* mice injected with corn oil, treated with EF-1 or Veh and corresponding graphs showing the quantitation for total and perivascular fibrosis. Scale bar = 500 μ m for total fibrosis, scale bar = 50 μ m for perivascular fibrosis. All data are represented as mean \pm SD, n = 3–5 in each group, one-way ANOVA with Tukey's multiple comparisons test. For the survival curve analysis, n = 6–15 in each group, Kaplan-Meier test with the log-rank (Mantel-Cox) test. AngII, angiotensin II; ANOVA, analysis of variance; CTRL, control; ERBB4, erythroblastic leukemia viral oncogene homolog 4; GAPDH, glyceraldehyde-3-phosphate dehydrogenase; SD, standard deviation; Tam, tamoxifen; Veh, vehicle."

Comment 10

In Figure 5a, the "B", "C", "D", and "E-G" should most likely not be capitalized. Same applies to Figure 6.

We now use lowercase letters, in line with the journal's guidelines.

Comment 11

In Figure 5e, total fibrosis is barely visible. Please provide close ups that clearly show an increase in interstitial fibrosis upon AngII treatment.

We have added close up images of the heart tissue within figure 5e.

Figure 5e

"Figure 5. EF-1 prevents cardiac fibrosis *in vivo*. (e) Representative images of Masson's trichrome staining of AngII-induced myocardial fibrosis following treatment with EF-1 or Veh and corresponding bar graphs showing the quantitation for total and perivascular fibrosis in male wild type mice."

Comment 12

Please note, that the role of NRG1/ERBB4 in heart regeneration PMID: 19632177 has been challenged PMID: 25848746.

We appreciate the reviewer bringing to our attention the evolving understanding of NRG1/ERBB4 signaling in cardiac regeneration. The study by Bersell et al. demonstrated a potential role for NRG1/ERBB4 in promoting cardiomyocyte proliferation and heart regeneration, generating excitement

around the therapeutic potential of this pathway.² However, we acknowledge that more recent research, such as the study by D'Uva et al.,³ raises important questions about the extent to which NRG1/ERBB4 contributes to cardiac repair, particularly in adult mammals. We included this reference in our manuscript.

Comment 13

Authors need to provide the approval number of their animal study.

The approval number was added to the methods section.

Methods

Page 15, lines 308-312

“All animal experiments were approved by the Ethical Committee of the University of Antwerp (approval number ECD 2020-08) and conformed to the Guide for the Care and Use of Laboratory Animals, 8th edition published by the US National Institutes of Health in 2011, and to the European Communities Council Directive 2010/63/EU for the protection of animals used for experimental purposes.”

Reviewer #3 comments:

This manuscript describes identification of small molecule activators of ErbB4 with a high throughput screen of a large library of compounds, and using a variety of in vitro and in vivo models demonstrates that the selected compound is able to recapitulate cardioprotective and antifibrotic effects of recombinant neuregulin-1. The potential for the NRG/ErbB signaling system to be a target of therapy in heart disease is not novel. However this represents a significant advance to this field as it represents the first demonstration of a small molecule activator of this pathway, which has therapeutic potential.

We thank the reviewer for their careful evaluation of our manuscript.

Comment 1

It appears that the small molecule selected is also able to induce ErbB3/ErbB2 heterodimer formation based upon a ErbB3/ErbB2 screen. Whether it also induces ErbB4/ErbB3 heterodimer formation is not specifically assessed, but seems likely given that it activates both ErbB3 and ErbB4 heterodimers with ErbB2. There is also no assessment of its effects on ErbB1 dimerization. Additional experimental work would be helpful to more completely understand the actions of the selected compounds.

We understand the reviewer's question regarding the potential dimerization effects of the small molecule selected in our study. Regarding the generation of potential ErbB4/ErbB3 dimers, there currently is no assay available that allows to assess ErbB4/ErbB3 dimerization.

As suggested by the reviewer, we studied the effect of our compound on ErbB1 (EGFR) dimerization and observed that EF-1 can induce Erbb1 dimerization at the highest concentration, but with an EC50 > 32 μ M.

Changes made to the manuscript: we included the results in Figure 1 of the revised manuscript to provide a more comprehensive understanding of the small molecule's actions. We also adapted the methods and results section.

Results

Page 4, lines 76-77

“We also noted that EF-1 activated ERBB1/ERBB1 dimerization, though this effect required higher concentrations (EC50 > 32 μ M, Fig. 1e).”

Methods

Page 15, lines 317-319

“PathHunter U2OS ERBB4/ERBB4 (Eurofins, 93-0961C3), U2OS ERBB2/ERBB4 (Eurofins, 93-0960C3), ERBB2/ERBB3 (Eurofins, 93-1042C3) and ERBB1/ERBB1 (Eurofins, 93-0989C3) dimerization cell lines were cultured according to the manufacturer’s instructions.”

Page 17, lines 367-368

“The same experimental set-up was used for the U2OS ERBB2/ERBB4, U2OS ERBB2/ERBB3 and U2OS ERBB1/ERBB1 dimerization cell lines.”

Figure 1

“Figure 1. High-throughput screening to identify small-molecule ERBB4 agonists. (b-e) DRC of the most potent compound EF-1 in (b) the ERBB4/ERBB4 dimerization assay, (c) the ERBB2/ERBB4 dimerization assay, (d) the ERBB2/ERBB3 dimerization assay and (e) the ERBB1/ERBB1 dimerization assay.”

Comment 2

While the ErbB4 silencing experiments and work with the ErbB4 knockout all support an activity in the selected assays that involves ErbB4, ErbB3 is also present in the cells, as well as the in vivo experimental systems selected. As the small molecule activator of ErbB4 also activates ErbB3, experimental studies addressing the effect of ErbB3 silencing would be needed to fully address the potential role of ErbB3 activation in these experimental systems, and fully support the conclusions as written throughout the manuscript.

The reviewer asks about the potential role of ErbB3 activation in the effects of EF-1. To further understand the role of ErbB3, we first measured the expression of ERBB3 in the cells used in our experiments. We used western blot to verify the expression of ErbB3 iAMs and cardiac fibroblasts and used hepatocytes as a positive control. In contrast to the expression in hepatocytes, we did not detect ErbB3 expression in iAMs nor in cardiac fibroblasts. Here we show the full western blot with the three different cell types.

Given the absence of significant levels of ERBB3 expression in iAMs and fibroblasts, we did not proceed with ErbB3 silencing experiments in the cells, and assume that the role of ErbB3 is neglectable.

Comment 3

It would be interesting to know if the library contained compounds active in the ErbB3/ErbB2 assay, but did not activate ErbB4. Study of such a compound in the select assays might serve as an interesting comparison.

We agree with the reviewer that exploring whether the library contains compounds that activate the ErbB3/ErbB2 assay but not ErbB4 could provide valuable comparative insights. However, this would require an additional high-throughput screening, which would be beyond the scope of our current study.

Comment 4

It was not clear whether the effect of gender was examined in the in vivo work. This would seem important as there have been some instances where the neuregulin/ErbB signaling system has distinct effects in male vs. female mice (e.g. H. Yin et al, JMCC 2021, PMID: 33259856).

We thank the reviewer for the relevant question about the role of sex in the effects of EF-1. To examine sex-related differences in the response to EF-1, we conducted additional *in vivo* experiments. Specifically, we included female mice in the angiotensin-II mouse model study, male mice in the acute toxicity study with doxorubicin and male mice in the myocardial infarction study. The results are described below.

Changes made to the manuscript: the results from the additional *in vivo* follow-up experiments are now included in the revised manuscript, with new data presented in Figure 5, Figure 6 and Figure 7 and discussed in the methods, results and discussion section. The echocardiography data were included in Table 1.

Results

Page 8, lines 163-186

“To evaluate changes in mRNA expression of markers of fibrosis, EF-1 was administered simultaneously with AngII for 1 week in male wild type mice (Fig. 5a). EF-1 not only significantly reduced *Col1a1* and *Col3a1* mRNA expression induced by AngII (Fig. 5b, 5c), indicating anti-fibrotic and cardioprotective properties, but also significantly decreased atrial natriuretic peptide (*Nppa*) expression (Fig. 5d), a cardiac stretch marker. To study effects on tissue fibrosis, EF-1 was administered simultaneously with AngII for 4 weeks in male and female wild type mice. Masson's trichrome staining of myocardial tissues showed that EF-1 significantly prevented both interstitial and perivascular fibrosis, induced by AngII (Fig. 5e; Fig. 5f). EF-1 did not significantly alter cardiac dimensions on ultrasound in both sexes (Suppl. Table 1a, 1b). To evaluate whether the observed anti-fibrotic effects could be mediated by targets unrelated to ERBB4, we performed a similar experiment with NA-1 (containing the common pharmacophore but without induction of ERBB4 dimerization) in male wild type mice. NA-1 did not significantly prevent interstitial or perivascular AngII-induced fibrosis (Suppl. Fig. 6a). Additionally, we evaluated the effects of EF-1 in Ang II-treated male and female transgenic mice with tamoxifen-induced deletion of *ErbB4* (*ErbB4*-null mice). *ErbB4* deletion was confirmed in the heart by western blot analysis (Suppl. Fig. 6b). As expected, *ErbB4* deletion resulted in a cardiomyopathy phenotype, with a significant increase in left ventricular internal diameter in diastole (LVIDd) and systole (LVIDs), and a decrease in

fractional shortening (FS) (Suppl. Table 1d). Since the commonly used AngII dose of 1,000 ng/kg/day induced 80% mortality in the *ErbB4*-null mice (Suppl. Fig. 6c), the dose of AngII was lowered to 400 ng/kg/day,¹⁵ EF-1 did not significantly prevent interstitial or perivascular AngII-induced fibrosis in *ErbB4*-null mice indicating that ERBB4 is necessary for the effects of EF-1 on cardiac fibrosis (Fig. 5g). Of note, when *iCAGCre-ErbB4^{fl/fl}* mice were injected with corn oil without tamoxifen, the inhibitory effects of EF-1 on AngII-induced interstitial and perivascular fibrosis were preserved (Suppl. Fig. 6d)."

Page 9, lines 190-197

"Osmotic minipumps with either EF-1 or Veh were implanted in male and female wild type and female *ErbB4*-null mice. Four days after the start of the EF-1 or Veh treatment, the animals received an intraperitoneal injection of 20 mg/kg DOX followed 3 days later by the measurement of cardiac troponin I (cTnI) plasma levels. EF-1 significantly prevented the DOX-induced increase in circulating cTnI plasma levels in female mice (Fig. 6b), but not in male mice (Fig. 6c). To verify whether the cardioprotective effects of EF-1 in female mice were mediated by ERBB4, we repeated the experiment using female *ErbB4*-null mice and observed that EF-1 did not prevent the increase in cTnI plasma levels in these mice (Fig. 6d)."

Pages 9-10, lines 199-220

"In view of the above effects induced by EF-1, and also because NRG1 has been shown to attenuate adverse cardiac remodeling in MI models, we assessed the effects of EF-1 in a murine MI model (Fig. 7a).⁵⁻⁷ Male and female mice were randomized 1 week after ligating the left anterior descending artery (LAD) to implantation of osmotic minipumps with either EF-1 or Veh. Treatment lasted for 28 days, during which cardiac size and function was evaluated using echocardiography (Suppl. Table 1e, 1f). Representative echocardiographic images of all groups in female mice are shown in Fig. 7b. In Veh-treated female mice, MI significantly increased left ventricular end-diastolic volume (LVEDV; Fig. 7c) and left ventricular end-systolic volume (LVESV; Fig. 7d) and decreased left ventricular ejection fraction (EF, Fig. 7e). Four-week treatment with EF-1 significantly attenuated the increase in LVEDV and LVESV and resulted in a trend towards an increased EF ($p = 0.0858$). In male mice, EF-1 did not have affect LVEDV, LVESV, or EF (Fig. 7c, 7d, 7e).

Hypertrophy and vascularization in heart tissue were visualized using an isolectin-B4/Wheat Germ Agglutinin (WGA)/DAPI staining. There was a trend to increase in cardiomyocyte cross-sectional area in the MI/Veh and MI/EF-1 group and a significant increase in cross-sectional area in the MI/EF-1 group, in both male and female mice (Fig. 7f). Additionally, a marked decrease in number of capillaries per area (μm^2) of the heart was detected in both MI/Veh and MI/EF-1 groups in female mice, but not in male mice (Fig. 7g). This decrease was not observed when the number of capillaries was normalized to cardiomyocytes, in both sexes (Fig. 7h).

Interstitial fibrosis in the remote myocardium of the infarcted hearts was significantly lower in female mice treated with EF-1, but not in male mice (Fig. 7i; 7j). EF-1 did not affect scar size in female or male mice (Fig. 7k)."

Discussion

Page 11, lines 227-230

"*In vivo*, EF-1 mitigated myocardial fibrosis in an AngII model of cardiac fibrosis in both male and female mice, prevented acute cardiomyocyte injury in DOX-treated female mice, and reduced cardiac dilation and cardiac fibrosis in female mice that underwent MI."

Pages 13-14, lines 288-299

“It is known that the NRG1/ERBB signaling system has distinct effects in male vs. female mice.¹⁶ For that reason, we investigated both male and female sexes in our *in vivo* experiments and we observed both shared and sex-specific responses across the three models of cardiac injury. In the angiotensin-II-induced myocardial fibrosis model, both male and female mice exhibited a significant reduction in total and perivascular fibrosis, suggesting that the intervention attenuates fibrotic remodeling in both sexes. However, in the doxorubicin-induced cardiotoxicity and myocardial infarction models, we observed a pronounced sex-specific effect, with only female mice showing significant improvements in cardiotoxicity and post-infarction heart function, respectively. These findings highlight potential sex differences in the response to cardiac injury, particularly in cardioprotection and recovery mechanisms, which may be influenced by hormonal or molecular factors unique to females. This underscores the importance of considering sex as a biological variable in preclinical studies and warrants further investigation.”

Methods

Page 24, lines 536-545

“Thirteen-week-old C57BL/6N (Charles River, 027) female and male mice were randomized to the ctrl group (n = 5 male mice, n = 8 female mice) or to the groups treated with EF-1 (EF-1 group, n = 4 male mice), AngII plus vehicle (AngII/Veh group; n = 5 male mice, n = 8 female mice), or AngII plus EF-1 (AngII/EF-1 group; n = 5 male mice, n = 8 female mice). AngII (1,000 ng/kg/min in PBS), Veh (DMSO/propylene glycol/50:50), and EF-1 (2 mg/kg/day in Veh) were administered for 4 weeks using subcutaneously implanted micro-osmotic pumps (Alzet, model 1004). Four weeks after implantation, cardiac ultrasound was performed, mice were euthanized, and hearts were collected. A similar set-up was used for the 1-week study (Alzet, model 1007D) in male mice (n = 10 mice in every group). In some experiments, EF-1 was replaced by NA-1 (2 mg/kg/day in Veh; n = 5 male mice in ctrl and AngII/Veh group, n = 4 male mice in AngII/EF-1 group).”

Page 24, lines 547-549

“Twelve-week-old C57BL/6N male and female mice were randomized to the ctrl group (n = 9 female mice, n = 8 male mice), or to the groups treated with DOX plus Veh (DOX/Veh group; n = 8 female mice, n = 9 male mice), or DOX plus EF-1 (DOX/EF-1 group; n = 8 female mice, n = 9 male mice).”

Page 25, lines 555-568

“Twelve-week-old female and male Balb/cJ mice were randomized into either the ctrl or MI group. Mice in the MI group underwent surgical ligation of the LAD, while mice in the ctrl group underwent a sham procedure (n = 8 female mice, n = 9 male mice). Briefly, all mice received an injection of 0.1 mg/kg buprenorphine (Produlab Pharma) before induction of anesthesia with 8% sevoflurane (Zoetis). Anesthesia was maintained with 4.5% sevoflurane and mice were intubated and ventilated. An incision of 15 mm was made on the left side of the thorax and the thoracic cavity was opened at the third intercostal space using blunt forceps. The LAD was permanently ligated with a 8/0 polypropylene monofilament sutures (Ethicon, F1894) after which the thoracic cavity and skin were closed with 6/0 polypropylene monofilament sutures (Ethicon, F1841). and 5/0 polyamide 6 sutures (Ethilon, F2412H), respectively. Mice received another dose of 0.1 mg/kg buprenorphine 6–8 h after the initial dose. Two days after surgery, cardiac ultrasound was performed to exclude mice without successful MI (i.e. displaying hypokinesia in ≥ 2 out of 5 segments). Mice with successful MI were randomized into 2

groups: an MI/EF-1 group (n = 9 female mice, n = 10 male mice) and an MI/Veh group (n = 8 female mice, n = 12 male mice).”

Page 26, lines 580-587

“Mice were used in aforementioned studies two weeks after the start of the tamoxifen injections (for the AngII-experiment with EF-1, n = 3 male and 3 female ctrl mice, n = 2 male and 3 female mice in the AngII/Veh group, n = 3 male and 2 female mice in the AngII/-EF-1 group; for the DOX-experiment, n = 8 female ctrl mice, n = 9 female DOX/Veh mice, n = 8 female mice in the DOX/EF-1 group; for the survival analysis, we used 3 male and 3 female ctrl mice and 9 male and 6 female AngII/Veh mice). As a control experiment, eleven-week-old female and male *iCAGGCre-ERTM/ErbB4^{fl/fl}* mice were injected with corn oil without tamoxifen (n = 4 male and 4 female ctrl mice, n = 5 male and 4 female mice in the AngII/Veh group and n = 5 male and 5 female mice in the AngII/EF-1 group).”

Figure 5

“Figure 5. EF-1 prevents cardiac fibrosis *in vivo*. (a) Design of the *in vivo* experiments. Hearts were collected after 1 week for mRNA analysis and after 4 weeks for histological analysis. **(b-d)** RT-qPCR was performed in male mice on fibrosis and cardiac stress markers; normalized FC compared to Ctrl: **(b)** *Col1a1*, **(c)** *Col3a1* and **(d)** *Nppa*. **(e)** Representative images of Masson’s trichrome staining of AngII-induced myocardial fibrosis following treatment with EF-1 or Veh and corresponding bar graphs showing the quantitation for total and perivascular fibrosis in male wild type mice. **(f)** Representative images of Masson’s trichrome staining of AngII-induced myocardial fibrosis following treatment with EF-1 or Veh and corresponding bar graphs showing the quantitation for total and perivascular fibrosis in female wild type mice. **(g)** Representative images of Masson’s trichrome staining of AngII-induced myocardial fibrosis in male and female *ErbB4-null* mice, treated with EF-1 or Veh and corresponding graphs showing the quantitation for total and perivascular fibrosis. Scale bar = 500 μ m for total fibrosis, 50 μ m for perivascular fibrosis. All data are represented as mean \pm SD, n = 4–8 in each group, one-way ANOVA with Tukey’s multiple comparisons test. AngII, angiotensin II; ANOVA, analysis of variance; Col1a1, collagen type 1 alpha 1; Col3a1, collagen type 3 alpha 1; Ctrl, control; echo, echocardiography; FC, fold change; NA, non-activating pharmacophore-containing compound; Nppa, atrial natriuretic peptide; RT-qPCR, reverse transcription-quantitative polymerase chain reaction; SD, standard deviation; Veh, vehicle.”

Figure 6

“Figure 6. EF-1 prevented cardiomyocyte cell death *in vivo* in female mice. (a) Design of the *in vivo* DOX experiment. (b-d) Graphs showing the cTnI levels measured in plasma samples of untreated control mice and of mice treated for 7 days with EF-1 or Veh and given a single IP injection of 20 mg/kg DOX on day 4 of treatment for (b) female wild type mice, (c) male wild type mice and (d) female *Erbb4*-null mice. All data are represented as mean \pm SD, n = 8–9 in each group, one-way ANOVA test with Tukey’s correction for multiple testing. ANOVA, analysis of variance; cTnI, cardiac troponin I; Ctrl, control; DOX, doxorubicin; IP, intraperitoneal; SD, standard deviation; Veh, vehicle.”

Figure 7

“Figure 7. EF-1 decreases ventricular dilation after MI in female mice. (a) Design of the *in vivo* MI experiment. **(b)** Representative echocardiographic images in end-diastole and end-systole of sham-operated control female mice and of mice that underwent MI for 4 weeks with EF-1 or Veh. Graphs showing **(c)** LVEDV, **(d)** LVESV and **(e)** EF in the different experimental groups during the entire experiment in both female and male mice. **(f)** CSA of cardiomyocytes analyzed with an isolectin-B4/WGA/DAPI staining in female and male mice. **(g-h)** Vascularization analysis of capillaries with an isolectin-B4/WGA/DAPI staining, showing the number of capillaries in female and male mice **(g)** per area of cardiomyocytes and **(h)** per 100 cardiomyocytes. **(i)** Representative images of Masson’s trichrome staining of interstitial fibrosis in the remote zone in female mice. **(j)** Corresponding graph quantifying the fibrotic area in the different experimental groups in female mice and the graph of the quantification of the fibrotic area in male mice. **(n)** Scar size analysis of the myocardial infarction area in female and male mice. Scale bar = 50 μ m. All data are represented as mean \pm SD, n = 8–10 in each group, one-way ANOVA test with Tukey’s correction for multiple testing. ANOVA, analysis of variance; CSA, cross-sectional area; Ctrl, control; DAPI, 4',6-Diamidino-2-phenylindole dihydrochloride; EF, ejection fraction; IP, intraperitoneal; LVEDV, left-ventricular end-diastolic volume; LVESV, left-ventricular end-systolic volume; MI, myocardial infarction; SD, standard deviation; Veh, vehicle; WGA, wheat germ agglutinin.”

a)	Male wild-type mice treated with AngII and EF-1			
	Ctrl	EF-1	AngII/Veh	AngII/EF-1
IVSd (mm)	0.9 ± 0.2	0.9 ± 0.1	0.8 ± 0.2	0.8 ± 0.1
IVSs (mm)	1.1 ± 0.2	1.2 ± 0.1	1.0 ± 0.1	1.1 ± 0.2
LVIDd (mm)	4.3 ± 0.4	4.2 ± 0.3	4.3 ± 0.3	4.4 ± 0.3
LVIDs (mm)	3.5 ± 0.4	3.4 ± 0.3	3.7 ± 0.4	3.7 ± 0.4
LVPWd (mm)	0.7 ± 0.1	0.75 ± 0.03	0.8 ± 0.1	0.8 ± 0.2
LVPWs (mm)	0.9 ± 0.1	0.9 ± 0.1	0.9 ± 0.2	0.9 ± 0.2
FS (%)	18.0 ± 2.9	18.5 ± 5.1	14.1 ± 4.0	16.6 ± 3.6
b)	Female wild-type mice treated with AngII and EF-1			
	Ctrl	AngII/Veh	AngII/EF-1	
IVSd (mm)	0.51 ± 0.08	0.5 ± 0.1	0.53 ± 0.07	
IVSs (mm)	0.7 ± 0.1	0.7 ± 0.2	0.7 ± 0.1	
LVIDd (mm)	4.6 ± 0.4	4.6 ± 0.3	4.2 ± 0.3	
LVIDs (mm)	3.9 ± 0.6	4.0 ± 0.3	3.7 ± 0.4	
LVPWd (mm)	0.49 ± 0.08	0.5 ± 0.1	0.5 ± 0.1	
LVPWs (mm)	0.7 ± 0.1	0.6 ± 0.1	0.6 ± 0.1	
FS (%)	15.7 ± 7.4	11.7 ± 6.2	11.5 ± 5.5	
c)	Male wild-type mice treated with AngII and NA-1			
	Ctrl	AngII/Veh	AngII/NA-1	
IVSd (mm)	0.8 ± 0.1	1.0 ± 0.2	1.0 ± 0.2	
IVSs (mm)	1.1 ± 0.1	1.3 ± 0.3	1.4 ± 0.2	
LVIDd (mm)	4.1 ± 0.4	4.3 ± 0.5	4.3 ± 0.7	
LVIDs (mm)	3.4 ± 0.4	3.5 ± 0.3	3.5 ± 0.7	
LVPWd (mm)	0.8 ± 0.3	1.2 ± 0.4	0.9 ± 0.1	
LVPWs (mm)	1.0 ± 0.3	1.4 ± 0.4	1.1 ± 0.3	
FS (%)	18.2 ± 2.0	18.8 ± 5.0	18.1 ± 5.1	
d)	Validation male and female ErbB4 -null mice			
	Wild-type mice	ErbB4 -null mice	P-value	
LVIDd (mm)	4.2 ± 0.4	5.0 ± 0.5	0.003	
LVIDs (mm)	3.5 ± 0.4	4.6 ± 0.6	0.0006	
FS (%)	18.1 ± 2.3	9.4 ± 6.1	0.001	
e)	Male and female ErbB4 -null mice treated with AngII and EF-1			
	Ctrl	AngII/Veh	AngII/EF-1	P-value
IVSd (mm)	0.9 ± 0.2	0.8 ± 0.2	1.0 ± 0.3	/
IVSs (mm)	1.1 ± 0.2	0.9 ± 0.3	1.2 ± 0.5	/
LVIDd (mm)	5.0 ± 0.5	5.5 ± 0.5	5.8 ± 0.3	‡0.04
LVIDs (mm)	4.6 ± 0.6	5.0 ± 0.7	5.5 ± 0.5	/
LVPWd (mm)	0.7 ± 0.1	0.8 ± 0.1	1.0 ± 0.4	‡0.05
LVPWs (mm)	0.7 ± 0.1	0.9 ± 0.1	1.1 ± 0.4	/
FS (%)	9.4 ± 6.1	8.3 ± 5.6	5.8 ± 3.5	/
f)	Myocardial infarction in female wild-type mice treated with EF-1			
	Ctrl	MI/Veh	MI/EF-1	P-value
T0				
LVEDA (mm ²)	21.3 ± 2.4	22.2 ± 2.8	21.8 ± 1.4	/
LVESA (mm ²)	14.8 ± 1.5	14.5 ± 1.4	15.5 ± 2.1	/
LVEDV (μL)	51.9 ± 7.9	57.1 ± 11.5	56.7 ± 5.5	/
LVESV (μL)	31.2 ± 6.2	27.9 ± 4.3	32.7 ± 6.7	/

EF (%)	42.7 ± 10.1	49.5 ± 11.5	42.7 ± 8.1	/
T2				
LVEDA (mm ²)	23.2 ± 2.1	27.2 ± 4.8	25.7 ± 1.6	¥0.04
LVESA (mm ²)	16.1 ± 1.7	22.3 ± 5.8	20.8 ± 2.9	¥0.01; †0.05
LVEDV (µL)	63.6 ± 9.9	83.8 ± 24.0	73.6 ± 9.1	¥0.04
LVESV (µL)	34.9 ± 7.0	62.0 ± 28.0	53.9 ± 14.5	¥0.01
EF (%)	44.6 ± 5.0	29.0 ± 14.8	27.4 ± 14.1	¥0.05; †0.02
T3				
LVEDA (mm ²)	21.6 ± 2.1	31.5 ± 6.0	26.5 ± 4.5	¥0.0005
LVESA (mm ²)	15.5 ± 2.1	28.0 ± 6.6	21.2 ± 5.8	¥0.0003; †0.03
LVEDV (µL)	55.1 ± 7.7	107.5 ± 29.1	80.3 ± 22.9	¥0.0002; †0.04
LVESV (µL)	31.1 ± 7.6	86.5 ± 32.3	56.6 ± 25.0	¥0.0002; †0.04
EF (%)	42.8 ± 6.3	21.5 ± 9.4	32.2 ± 12.4	¥0.0007
T4				
LVEDA (mm ²)	21.6 ± 2.1	31.5 ± 6.0	26.5 ± 4.5	¥0.0005
LVESA (mm ²)	15.5 ± 2.1	28.0 ± 6.6	21.2 ± 5.8	¥0.0003; †0.03
LVEDV (µL)	55.1 ± 7.7	107.5 ± 29.1	80.3 ± 22.9	¥0.0002; †0.04
LVESV (µL)	31.1 ± 7.6	86.5 ± 32.3	56.6 ± 25.0	¥0.0002; †0.04
EF (%)	42.8 ± 6.3	21.5 ± 9.4	32.2 ± 12.4	¥0.0007
g)	Myocardial infarction in male wild-type mice treated with EF-1			
	Ctrl	MI/Veh	MI/EF-1	P-value
T0				
LVEDA (mm ²)	24.1 ± 2.8	25.7 ± 2.3	24.7 ± 3.9	/
LVESA (mm ²)	17.7 ± 2.6	18.4 ± 3.1	17.6 ± 2.8	/
LVEDV (µL)	63.0 ± 11.6	71.4 ± 13.2	67.1 ± 15.6	/
LVESV (µL)	40.0 ± 9.9	40.5 ± 13.1	37.3 ± 8.9	/
EF (%)	41.2 ± 8.8	43.5 ± 11.1	43.6 ± 9.5	/
T2				
LVEDA (mm ²)	26.2 ± 2.1	34.8 ± 5.1	34.8 ± 5.5	¥0.0007; †0.001
LVESA (mm ²)	18.9 ± 2.2	28.9 ± 5.2	29.7 ± 6.7	¥0.0004; †0.0003
LVEDV (µL)	72.7 ± 8.8	117.7 ± 28.6	118.7 ± 30.8	¥0.001; †0.001
LVESV (µL)	41.9 ± 7.5	89.0 ± 27.3	93.3 ± 36.1	¥0.001; †0.0008
EF (%)	42.2 ± 7.7	24.9 ± 11.2	23.2 ± 12.8	¥0.003; †0.002
T3				
LVEDA (mm ²)	23.4 ± 2.2	31.2 ± 4.0	36.6 ± 5.9	¥0.02; †0.0008
LVESA (mm ²)	15.5 ± 2.7	25.0 ± 5.7	31.3 ± 8.0	¥0.05; †0.003
LVEDV (µL)	61.2 ± 7.6	95.1 ± 17.1	124.6 ± 35.3	†0.002
LVESV (µL)	30.2 ± 7.3	67.9 ± 26.2	100.4 ± 42.0	†0.005
EF (%)	50.5 ± 10.2	30.7 ± 15.2	22.0 ± 15.7	†0.02
T4				
LVEDA (mm ²)	26.3 ± 2.5	33.2 ± 6.1	35.9 ± 6.9	¥0.02; †0.002
LVESA (mm ²)	18.4 ± 2.3	26.2 ± 3.7	30.5 ± 7.7	¥0.006; †<0.0001
LVEDV (µL)	72.7 ± 12.4	108.8 ± 36.8	119.8 ± 37.5	¥0.04; †0.009
LVESV (µL)	42.3 ± 9.9	72.3 ± 18.5	92.5 ± 40.6	¥0.05; †0.009
EF (%)	42.0 ± 7.2	26.0 ± 10.5	25.2 ± 11.5	¥0.003; †0.003

“Supplementary Table 1. Echocardiographic parameters. Echocardiographic parameters obtained through the analysis of the M-mode short-axis views. **(a)** Male and **(b)** female wild-type mice treated with AngII and EF-1. **(c)** Male wild-type mice treated with AngII and NA-1. **(d)** Comparison between wild-type and *ErbB4-null* mice. **(e)** Male and female *ErbB4-null* mice treated with AngII and EF-1. **(f)** Echocardiographic parameters obtained through left ventricular trace analysis of the parasternal long-axis obtained in B-mode in female wild-type mice that underwent a sham or MI operation: mice with the MI were treated with Veh or EF-1. **(g)** Echocardiographic parameters obtained through left ventricular trace analysis of the parasternal long-axis obtained in B-mode in male wild-type mice that underwent a sham or MI operation: mice with the MI were treated with Veh or EF-1. Echocardiographic parameters were measured weekly (indicated by ‘t’). All data are presented as mean \pm SD, n = 5-10 in each group, one-way ANOVA test with Tukey’s correction for multiple testing. \yen shows significant P-values between **(f-g)** ctrl and MI/Veh groups; \ddagger shows significant P-values between **(e)** ctrl and AngII/EF-1 groups and **(f-g)** ctrl and MI/EF-1 groups; \dagger shows significant P-values between **(f-g)** MI/Veh and MI/EF-1 groups. AngII, angiotensin II; ANOVA, analysis of variance; Ctrl, control; DOX, doxorubicin; EF, ejection fraction; ERBB4, erythroblastic leukemia viral oncogene homolog 4; FS, fractional shortening; IVSd, interventricular septum thickness in diastole; IVSs, interventricular septum thickness in systole; LVIDd, left ventricular internal diameter in diastole; LVIDs, left ventricular internal diameter in systole; LVPWd, left ventricular posterior wall thickness in diastole; LVPWs, left ventricular posterior wall thickness in systole; MI, myocardial infarction; NA, non-activating pharmacophore-containing compound; SD, standard variation; Veh, vehicle.”

References

This reference list is automatically generated with references in this document and thus the order is different from the order in the revised manuscript.

1. Wang, F. *et al.* Pharmacological postconditioning with Neuregulin-1 mimics the cardioprotective effects of ischaemic postconditioning via ErbB4-dependent activation of reperfusion injury salvage kinase pathway. *Molecular Medicine* 24, (2018).
2. Bersell, K., Arab, S., Haring, B. & Kühn, B. Neuregulin1/ErbB4 Signaling Induces Cardiomyocyte Proliferation and Repair of Heart Injury. *Cell* 138, 257–270 (2009).
3. D’Uva, G. *et al.* ERBB2 triggers mammalian heart regeneration by promoting cardiomyocyte dedifferentiation and proliferation. *Nat Cell Biol* 17, 627–38 (2015).
4. Reuter, S., Soonpaa, M. H., Firulli, A. B., Chang, A. N. & Field, L. J. Recombinant Neuregulin 1 Does Not Activate Cardiomyocyte DNA Synthesis in Normal or Infarcted Adult Mice. *PLoS One* 9, e115871 (2014).
5. Shiraishi, M., Yamaguchi, A. & Suzuki, K. Nrg1/ErbB signaling-mediated regulation of fibrosis after myocardial infarction. *FASEB J* 36, e22150 (2022).
6. Hedhli, N. *et al.* Endothelium-Derived Neuregulin Protects the Heart Against Ischemic Injury. *Circulation* 123, 2254–2262 (2011).
7. Lin, Y., Liu, H. & Wang, X. Neuregulin-1, a microvascular endothelial-derived protein, protects against myocardial ischemia-reperfusion injury (Review). *Int J Mol Med* 46, 925–935 (2020).
8. Fang, S.-J. *et al.* Neuregulin-1 preconditioning protects the heart against ischemia/reperfusion injury through a PI3K/Akt-dependent mechanism. *Chin Med J (Engl)* 123, 3597–604 (2010).
9. Jie, B. *et al.* Neuregulin-1 suppresses cardiomyocyte apoptosis by activating PI3K/Akt and inhibiting mitochondrial permeability transition pore. *Mol Cell Biochem* 370, 35–43 (2012).
10. Gallo, S., Vitacolonna, A., Bonzano, A., Comoglio, P. & Crepaldi, T. ERK: A key player in the pathophysiology of cardiac hypertrophy. *Int J Mol Sci* 20, (2019).
11. Liu, J. *et al.* Generation and primary characterization of iAM-1, a versatile new line of conditionally immortalized atrial myocytes with preserved cardiomyogenic differentiation capacity. *Cardiovasc Res* 114, 1848 (2018).
12. Pentassuglia, L. *et al.* Neuregulin-1 β promotes glucose uptake via PI3K/Akt in neonatal rat cardiomyocytes. *Am J Physiol Endocrinol Metab* 310, E782-94 (2016).
13. Watkins, S. J., Borthwick, G. M. & Arthur, H. M. The H9C2 cell line and primary neonatal cardiomyocyte cells show similar hypertrophic responses in vitro. *In Vitro Cell Dev Biol Anim* 47, 125–131 (2011).
14. Watkins, S. J., Borthwick, G. M., Oakenfull, R., Robson, A. & Arthur, H. M. Angiotensin II-induced cardiomyocyte hypertrophy in vitro is TAK1-dependent and Smad2/3-independent. *Hypertension Research* 2012 35:4 35, 393–398 (2011).

15. Shakeri, H. *et al.* Neuregulin-1 compensates for endothelial nitric oxide synthase deficiency. *Am J Physiol Heart Circ Physiol* 320, H2416–H2428 (2021).
16. Yin, H. *et al.* Protective role of ErbB3 signaling in myeloid cells during adaptation to cardiac pressure overload. *J Mol Cell Cardiol* 152, 1–16 (2021).